# Design and Synthesis of New Pyrimidine-Quinolone Hybrids as Novel *h*LDHA Inhibitors

**DOI:** 10.3390/ph15070792

**Published:** 2022-06-24

**Authors:** Iván Díaz, Sofia Salido, Manuel Nogueras, Justo Cobo

**Affiliations:** Facultad de Ciencias Experimentales, Departamento de Química Inorgánica y Orgánica, Universidad de Jaén, E-23071 Jaén, Spain; idc00005@red.ujaen.es (I.D.); ssalido@ujaen.es (S.S.); mmontiel@ujaen.es (M.N.)

**Keywords:** *h*LDHA inhibitors, quinolones, pyrimidines, fragment-based drug design, docking

## Abstract

A battery of novel pyrimidine-quinolone hybrids was designed by docking scaffold replacement as lactate dehydrogenase A (*h*LDHA) inhibitors. Structures with different linkers between the pyrimidine and quinolone scaffolds (**10**-**21** and **24**–**31**) were studied in silico, and those with the 2-aminophenylsulfide (U-shaped) and 4-aminophenylsulfide linkers (**24**–**31**) were finally selected. These new pyrimidine-quinolone hybrids (**24**–**31**)(**a**–**c**) were easily synthesized in good to excellent yields by a green catalyst-free microwave-assisted aromatic nucleophilic substitution reaction between 3-(((2/4-aminophenyl)thio)methyl)quinolin-2(1*H*)-ones **22/23**(**a**–**c**) and 4-aryl-2-chloropyrimidines (**1**–**4**). The inhibitory activity against *h*LDHA of the synthesized hybrids was evaluated, resulting IC_50_ values of the U-shaped hybrids **24**–**27**(**a**–**c**) much better than the ones of the 1,4-linked hybrids **28**–**31**(**a**–**c**). From these results, a preliminary structure–activity relationship (SAR) was established, which enabled the design of novel 1,3-linked pyrimidine-quinolone hybrids (**33**–**36**)(**a**–**c**). Compounds **35**(**a**–**c**), the most promising ones, were synthesized and evaluated, fitting the experimental results with the predictions from docking analysis. In this way, we obtained novel pyrimidine-quinolone hybrids (**25a**, **25b,** and **35a**) with good IC_50_ values (<20 μM) and developed a preliminary SAR.

## 1. Introduction

One of the main diseases that cause death and, therefore, one of the main public health problems worldwide continues to be cancer [1,2]. In the last decades, most of the main hallmarks of many cancers have been established [3]. In the case of metabolism alteration, in normal cells, glucose is metabolized into pyruvate and afterwards into carbon dioxide and acetyl-CoA through an oxidative phosphorylation process. In tumor cells, this process is highly disordered, as anaerobic glycolysis is often preferred over oxidative phosphorylation. This metabolic switch is known as the Warburg effect and leads to the formation of lactate [4]. In this switch, several studies suggest that lactate dehydrogenase A (*h*LDHA) enzyme plays a key role in cancer proliferation, as it is responsible for catalyzing the conversion of pyruvate into lactate [5,6,7,8,9].

Recently, *h*LDHA has also been shown to be implicated in other diseases such as primary hyperoxaluria (PH), which converts glyoxylate into oxalate [10,11]. When oxalate is overproduced, calcium oxalate crystals appear in the kidney, leading to urolithiasis, nephroncalcinosis, renal failure [12], and, eventually, end-stage renal disease [13,14,15]. Consequently, the *h*LDHA enzyme is an ideal therapeutic target for cancer and PH treatment.

The development of new chemical entities (NCEs) based on small molecules wearing aza-heterocyclic nuclei still constitute one of the most important areas within the pharmaceutical industry [16]. Those systems can be found in a huge range of drugs and bioactive compounds due to the fact they are the main pharmacophoric residues responsible for their biological response and/or for being the key synthetic scaffold, which is the case of pyrimidines and quinolones.

In particular, pyrimidine derivatives have shown diverse activities, such as antimicrobial, antioxidant, antimalarial, and anti-inflammatory [17]. Furthermore, they have been used as potential agents in the treatment of neurodegenerative diseases such as Alzheimer’s [18] and in the treatment of cancer [17,19,20,21,22]. Thus, pyrimidine, as a biologically privileged scaffold, is commonly used in the development of new drugs towards different targets [23,24].

Quinolones are also considered to be biologically privileged, as they interact with a diverse biotargets and show a wide variety of bioactivities, such as antiviral, anti-parasitic [25], anti-malarial [26], or anti-inflammatory [27] activities, amongst many others. They are also used as biomarkers [28] in the treatment of different types of cancer, as only heteronucleus [29,30], or in combination with other different scaffolds [31], such as benzo[*d*]thiazolyl [32], cinnamic acid [33], or with hydantoin searching for antimicrobial activity [34].

The hybridization of both systems, in accordance with the Fragment-Based Drug Discovery (FBDD) strategy [35,36,37,38], has been demonstrated to be highly interesting regarding their antiproliferative action [39], such as anticancer [40], anti-HIV [41], and antimalarial/antiplasmodial [26,42,43,44,45,46,47,48], and for being inhibitors of human sphingomyelin synthase 2 [49].

Some *h*LDHA inhibitors wearing the pyrimidine (**I–III**) and quinolone (**IV**) nucleus have already been reported [50,51]. However, we have only found a few examples of structure-related hybrids that have been reported as *h*LDHA inhibitors (**V** and **VI**) [52,53,54] but without linkers between the pyrimidine and quinoline fragments (Figure 1).

Another structural feature observed after a thorough analysis of different *h*LDHA inhibitors is that most of them (**VII–XII** in Figure 2) had a hydrophilic scaffold and a hydrophobic one, with or without a linking moiety separating them [4,5].

In that sense, we have already reported the synthesis of diverse hybrids bearing the quinolone fragment as potential antimalarial and antitumoral agents [55,56,57] and, amongst them, some with pyrimidine residues, which contain both the hydrophobic and hydrophilic scaffolds, shown to be promising anticancer agents [58,59]. In particular, we have recently reported the synthesis of a series of pyrimidine-quinolone hybrids following a linear synthetic methodology starting from 2,4-dichloropyrimidine (Figure 3) and proved their bioactivity as sphingosine kinase (SphK) inhibitors, which are involved in cell proliferation [58] and P-glycoprotein (P-gp) inhibitors in the search for reversal agents of multidrug resistance [59].

Bearing all this in mind, and taking these hybrids as the starting point for the development of a novel family of *h*LDHA inhibitors, we here report their rational design, synthesis, and biological evaluation. These NCEs are based on pyrimidine-quinolone hybrids linked by an aminophenylsulfide fragment in a U- and non-U-shaped disposition, which are of potential interest regarding their behavior as *h*LDHA inhibitors according to what has been mentioned previously.

## 2. Results and Discussion

### 2.1. Virtual Screening Scaffold Replacement in the Optimization of Pyrimidine-Quinolone Hybrids as hLDHA Inhibitors

Complex *h*LDHA-W31 (code **4R68**) was selected and downloaded from the Protein Data Bank (PDB) as reference for the docking studies due to the following reasons: (i) its ligand (**W31**) interacts with the main amino acid residues reported to be responsible for its activity (Arg^168^, Asn^137^, His^192^, and Asp^194^) [60], (ii) it occupies the whole substrate (pyruvate) pocket [61], and (iii) it has an IC_50_ = 6 nM [62].

The Figure 4a represents the **W31** placement in the substrate pocket and Figure 4b its 2-D interaction diagram with the main amino acid residues in that active site Blue spheres in left image represent the pharmacophore descriptor by imposed features where a hydrogen donor/acceptor atom could be located to interact with such key residues.

Firstly, we proceeded by excluding through docking screening any possibility of NADH competitive inhibition. Thus, in order to discard any other possible interaction sites of the pyrimidine-quinolone hybrids deigned in this work besides the expected **W31** site, we ran the docking process in triplicate with different docking areas and pharmacophoric descriptors [63] as described in Section 3.4: (i) in the *h*LDHA active site (**W31** site), (ii) in the NADH site, and (iii) in the extension covering both sites.

In that regard, based on our previous experience in the synthesis of pyrimidine-quinolone compounds [58], a first set of compounds (**10**–**21**) was designed, having the quinolone scaffold as the hydrophobic moiety and the 4-chlorophenyl scaffold as the hydrophobic one (Figure 5).

Compounds **10**–**12** were already synthesized by us and evaluated as sphingosine kinase inhibitors [58]. Structures **13**–**21**, with new linking precursors (1,3-diaminobenzene, 1,2-diaminobenzene, aminophenol, catechol, ethylenediamine, and ethanolamine), were designed for their in silico study.

The docking results showed that the inhibition is unlikely to take place by displacement of the NADH cofactor in its site, as the affinity values are not close enough to compete against it. This is reinforced by the fact that they do not give any interaction with those amino acid residues that interact strongly with NADH in its site. This way, the affinity and energy values involved in the interactions with the mentioned key amino acids in the **W31** site suggested that the inhibition may take place in such an *h*LDHA active site (see Appendix A).

Once the docking analysis was focused in the **W31** site, we proceeded with its deep analysis to determine the best poses for each ligand. We proceeded to filter them in the following order [64,65]: first, according to root mean square difference score (RMSD < 1.8 Å); second, after the refinement of the pose using molecular mechanics; and afterwards, according to affinity value (S < −9 kcal/mol) and then those showing interactions with key Arg^168^. Finally, the energy values involved in their interactions with the other key amino acid residues were compared.

After this filtering process, the docking results yielded a low affinity for structures **20** and **21** with an ethylene chain in the linker, and so, they did not overcome this filter criteria to pass the next level to check the interaction energies. Compounds **11**, **12,** and **16**–**18** did not afford any interaction with the key Arg^168^, and thus, they were not considered for the last filtering step. Only compounds **10, 13**–**15**, and **19** succeeded this screening.

When synthesizing the suggested hybrids, some difficulties were faced (see Section 2.2), which forced us to accomplish tiny modifications in the linking fragment.

Considering that **W31** ligand has a thio-substituted moiety, we postulated to exchange the oxygen atom for sulfur in such aminophenol linker. Therefore, the new structure (**24a**) redefined with the 2-aminothiophenol linker gives a slightly better affinity (−9.24 Kcal/mol) than some of those previously tested (**10**, **15,** and **19**) and similar to **13** and **14**. Additionally, **24a** shows interactions with two of the main amino acid residues (Arg^168^ and Asp^194^) as displayed in Figure 6. It is worth mentioning that this modification will also result in benefits during the synthetic stage.

At this point, we proceeded to extend the docking screening to a bigger battery of different pyrimidine-quinolone hybrids, regarding substitution in the designated hydrophilic and hydrophobic residues and also substitution at linker 1,2-linked (**24**–**27**)(**a**–**c**) and 1,4-linked (**28**–**31**)(**a**–**c**) (Figure 7).

After running the docking screening as above described, hybrids **24**–**27**(**a**–**c**) showed promising in silico results, with the 1,2-substitution at linker having much better affinity and energy values than compounds **28**–**31**(**a**–**c**) with the 1,4-substitution. Table 1 summarizes the docking results, reporting the mean energy and affinity data for each family regarding linker substitution. This way, hybrids **28**–**31**(**a**–**c**) do not show any interaction with other amino acid residues apart from Arg^168^ and slightly with Asn^137^. On the contrary, compounds **24**–**27**(**a**–**c**) do interact strongly not only with Arg^168^ but also with His^192^. They also show interactions with Asp^194^and Asn^137^.

To determine the effect of the aryl group attached to the pyrimidine nucleus, the so-called hydrophobic scaffold, within hybrids **24**–**27,** we proceeded similarly as described above, and the selected mean data are displayed in Table 2.

As it can be deduced from Table 2, hybrids **26**(**a**–**c**), with the naphthalen-2-yl moiety at pyrimidine, are expected to be the most interesting ones in order to inhibit the *h*LDHA enzyme, as they have the highest affinity value, and they show a very strong interaction with Arg^168^, which are the prime filtering criteria.

The higher affinity of derivatives **26**(**a**–**c**) is related to their better placement in the active site, as their hydrophobic naphthalen-2-yl moiety fits well in the lipophilic area of the active site (Figure 8).

### 2.2. Chemistry

In order to succeed in our first aim of synthesizing **13**–**15** and **19**, we first tried to benefit from our reported linear synthetic pathway based on the sequential introduction of fragments from 2,4-dichloropyrimidine [58].

Attempts to synthesize **13** resulted in extreme difficulties related to over-reactivity and, as a result, making the obtention of the mono-substituted intermediate almost impossible. This led us to discard that structure as well as its analogue, **14**.

To prepare hybrid **15** by that methodology, intermediate **5**, prepared from 4-aminophenol as linker precursor [58], was reacted with 3-bromomethylquinolin-2(1*H*)one **6a** (Figure 1), but this classic nucleophilic substitution did not work in any way tried. We proved a range of solvents from protic (EtOH) to polar aprotic (DMSO, ACN, DMF) or apolar (THF) and in combination with different bases (K_2_CO_3_, Et_3_N, NaH), but decomposition, or solvolysis in the case of EtOH, resulted. In turn, we made a detour and performed the nucleophilic substitution between **5** and the 3-bromomethyl-2-chloroquinoline **7** to give intermediate quinoline derivative **8** in 64%, which, after a further hydrolysis and heating in aqueous acetic acid solution, afforded the desired compound **15** in 61% (Figure 1).

Both compounds **8** and **15** were completely characterized by the standard spectroscopic and analytical methods. Hence, all the characteristic NMR signals corresponding the different aryl residues are found in both structures as well as the proper masses found in both HRMS and MS, in which is clearly observed the difference in the isotopic pattern for the two chlorine atoms in **8** with respect to one in **15**. The main difference in their ^1^H-NMR spectra is related to the change in quinoline residue because of the hydrolysis and loss of chlorine, resulting in the signal of the NH of the lactam-related structure at 11.99 ppm for **15**, which is not observed in **8**, and also the corresponding lactam C=O that now results for **15** both in ^13^C-RMN at 160.9 ppm and in its IR spectrum at 1661 cm^−1^.

Once compound **15** was synthesized, we found out that it was highly insoluble, which translated into a difficulty in measuring its inhibitory activity.

For the synthesis of **19,** we started from 1,2-dihydroxybenzene (catechol) by following a similar linear synthetic pathway to the one shown in Figure 1a for the obtention of **15**, but we did not obtain any reaction. A considerable number of attempts were tried in the last reaction step by using different bases (K_2_CO_3_, DIPEA, *t*-BuOK, and NaH), different conditions (room temperature, conventional heating, and microwave irradiation), different solvents from polar protic (EtOH, *t*-BuOH) to polar aprotic (DMF, DMSO, ACN) or slightly polar (THF), as well as silver nitrate as a catalyst, but none of them afforded the expected **19**.

To overcome the lack of reactivity of the free hydroxyl group when the catechol moiety is linked to the pyrimidine core, we first connected the catechol linker to the quinolone scaffold. Intermediate **9**, formed by reaction of catechol with **6a**, was reacted with 2-chloropyrimidine **1** to give the desired **19** (Figure 2) in a reasonably good yield of 67% in hot DMSO, using potassium carbonate as base and silver nitrate as catalyst.

Both the intermediate **9** and final product **19** were completely characterized. The reaction monitoring was performed by following in ^1^H-NMR spectrum the disappearing of the signal at 9.21 ppm belonging to the free hydroxyl group in **9** and the change in the chemical shift concerning the methylene moiety (from 4.98 ppm in **9** to 5.08 ppm in **19**).

After having had problems related with solubility (**10** and **15**) and reactivity (**13** and **19**), we decided to evaluate a slightly modified linker: 2-aminothiophenol. This way, after having studied in silico the benefits of this new linker as previously mentioned in Section 2.1 with structure **24a**, we dealt with the synthesis of hybrids **24**–**31** with 2/4-aminothiophenol as linker precursors.

Both methodologies (linear and convergent) were used to obtain **24a** as the final product, and only the latter convergent one, shown in Figure 2, succeeded. The synthesis of intermediate **22a** was optimized, and 2-aminothiophenol was reacted with 3-bromomethylquinolone **6a** at room temperature with a green solvent (ethanol) under the presence of potassium carbonate as a base.

For the last step, to afford the final hybrid **24a** from **22a** and **1**, the optimization of the reaction was made by two different heating methodologies:Under conventional heating (at reflux). Different polar solvents were tested, and after eight days, the reaction was not finished when ethanol was used. In order to increase reaction temperature, *n*-butanol was used, after which the reaction took more than eight days to complete but with a great deal of by-products;Under microwave irradiation. Using ethanol, the reaction time was drastically reduced to 15 min, which allowed us to synthesize the desired hybrid **24a** in 86% yield.

Following that convergent synthetic pathway under microwave irradiation, we managed to succeed in the synthesis of all the designed pyrimidine-quinolone hybrids **24**–**31**(**a**–**c**) in a straightforward manner (Figure 3), allowing us to corroborate the reliability of the previous in silico predictions. Reaction time and yields are indicated in Table 3.

Nonetheless, due to the drop in the reaction yield and higher reaction times in some cases, we tried to make some improvements in the methodology, but they were not achieved (see Appendix A). From all the attempts carried, the vast majority of them ended in the same way: we did not find reaction, and if a reaction did happen, the result was an extremely high number of by-products and decomposition of intermediate **22**–**23**.

An explanation for the fact that compounds **28**–**31**(**a**–**c**) showed higher yields than **24**–**27**(**a**–**c**) might be found in the larger steric hindrance between fragments around the linker in the latter 1,2-linked, which is not found in the case of the former 1,4-linked.

All the pyrimidine-quinolone hybrids **24**–**31**(**a**–**c**) shown in Figure 3 as well as intermediates **22**(**a**–**c**) and **23**(**a**–**c**) were completely characterized using the standard spectroscopic and analytical methods. We found remarkable the disappearance in the ^1^H-NMR spectra of the signal corresponding to the hydrogens of the primary amine at 5.30–5.70 ppm (belonging to -NH_2_) of **22**(**a**–**c**) and **23**(**a**–**c**) and the appearance of a new one between 8–9 ppm (belonging to the hydrogen of the secondary amine linked to C2 at pyrimidine) of the hybrids **24**–**27**(**a**–**c**), which is key to ensure the reaction has been produced.

IR spectra of intermediates **22**(**a**–**c**) and **23**(**a**–**c**) showed a double band at ≈3400 and ≈3300 cm^−1^ belonging to the asymmetric and symmetric stretching of the primary amine, respectively. Meanwhile, the final pyrimidine-quinolone hybrids **24**–**31**(**a**–**c**) showed only one band at ≈3200 cm^−1^, belonging to the N-H stretching for the secondary amine. In addition to this, a wide signal between 3500 and 2100 cm^−1^ appeared for both intermediates **22**–**23**(**a**–**c**) and hybrids **24**–**31**(**a**–**c**), which is typical of the NH for the lactam-related of the quinolone scaffold.

For compound **24b**, single crystals were obtained from DMSO, which allowed us to unambiguously corroborate its structure by single crystal X-ray diffraction (see Figure 9), which agrees with the spectroscopic characterization.

### 2.3. hLDHA Inhibitory Assays and Structure–Activity Relationship (SAR)

The inhibitory activity of the pyrimidine-quinolone hybrids **10**, **15**, **19,** and **24**–**31**(**a**–**c**) against the *h*LDHA enzyme was measured by a kinetic spectrofluorometric assay [66].

The first set of compounds (**10, 15,** and **19**) did not show good inhibitory activity, as their IC_50_ was >100 μM. Thus, in concordance with the docking results for the second set of hybrids **24**–**31**(**a**–**c**), the inhibitory activity of both 1,2-linked **24**–**27**(**a**–**c**) and 1,4-linked **28**–**31**(**a**–**c**) was measured.

1,2-Linked hybrids **24**–**27**(**a**–**c**), with the exception of **27a**, have IC_50_ values under 100 μM, from which seven have IC_50_ < 50 μM (**24a**, **24b**, **25a**, **25b**, and **26**(**a**–**c**)). However, from those 1,4-linked hybrids, only **29b**, **30a**, and **31b** have IC_50_ < 100 μM, with their values ranging between 50 and 83 μM (Table 4).

Amongst those 1,2-linked hybrids having IC_50_ < 50 μM, compounds **26**(**a**–**c**), having the napthalen2-yl moiety, are the ones with the best inhibitory results (as previously predicted), their IC_50_ values being 17.8, 20.3, and 27.7 μM, respectively.

1,4-Linked hybrids **28**–**31**(**a**–**c**) were predicted to be inactive; however, despite **31a** and **31c** being inactive, compound **31b** demonstrated an interesting IC_50_ = 49.9 μM as the only 1,4-linked hybrid with interesting inhibitory activity.

An explanation for that might be found in the placement of **31b** in the active site. Thus, meanwhile, **31a** did not even pass the filtering criteria, and **31c** had a very different placement to that of **W31**, and **31b** had a more similar one to **W31**, enabling some interactions with the different amino acid residues (Figure 10).

If we compare the inhibitory activity of all the 1,4-linked hybrids **28**–**31**(**a**–**c**) with the inhibitory activity of those 1,2-linked **24**–**27**(**a**–**c**), there is a correlation with the in silico studies, marking the importance of the U-shaped disposition to mimic the shaping of the reference **W31**.

The correlation found between the in silico studies and the experimental data encouraged us to design a preliminary structure–activity relationship. In this regard, we envisioned that perhaps the 1,3-linked pyrimidine-quinolone hybrids **33**–**36**(**a**–**c**) (Figure 11) may also be of interest, and we decided to study them in silico following the process described for the 1,2- and 1,4-linked pyrimidine-quinolone hybrids **24**–**31**(**a**–**c**) (Section 2.1).

Concerning to the affinity criteria, we found a tendency concerning the linker substitution where, when going from the 1,4-subtitution towards the 1,2-substituion, the affinity improved considerably as seen in Figure 12. In this figure, mean affinity values (kcal/mol) are represented grouped by linker families. Values were obtained from the minimization process made after obtaining the docking output file (see Appendix A).

Moreover, we found that there is also a clear relationship between the substitution pattern and the energy values involved in the interaction with the main amino acid residues (Figure 13). This way, all the structures evaluated show a strong interaction with key Arg^168^.

However, when considering the other amino acid residues (His^192^, Asn^137^, and especially with Asp^194^), there are some major differences. On the one hand, interactions with Asn^137^ and His^192^ do not enable a major difference in terms of defining whether the structure is a potential inhibitor or not, as their values are quite similar and around −1 kcal/mol.

On the other hand, and most importantly, interactions with Asp^194^ are an essential factor to discriminate, as those 1,4-linked hybrids (**28**–**31**)(**a**–**c**) do not interact with it, but those having the 1,2-substituition pattern (**24**–**27**)(**a**–**c**) demonstrate a strong interaction.

Now, after realizing the importance of the affinity and energy values involved in the interactions with the different amino acid residues, the fact that those compounds having the 4-chlorophenyl scaffold **24**(**a**–**c**) have similar IC_50_ values to those having the 4-trifluorophenyl **25**(**a**–**c**), respectively, is explained in silico by striking a balance between the better affinity values of **25**(**a**–**c**) and the better energy values involved in the interactions with Arg^168^ and Asp^194^ of **24**(**a**–**c**). Moreover, the absence of interactions in the 1,2-linked hybrids with the benzo[*d*][1,3]dioxol-5-yl moiety **27**(**a**–**c**) with Asp^194^ may explain the fact that they do not demonstrate high inhibitory activity despite having similar affinity values to the other 1,2-linked hybrids (see Table 2 in Section 2.1).

In light of the analysis of the data from the SAR, we realized that those 1,3-linked pyrimidine-quinolone hybrids (**33**–**36**)(**a**–**c**) might have interesting inhibitory activity, which is predicted to be between the 1,4-linked (**28**–**31**)(**a**–**c**) and 1,2-linked pyrimidine quinolone hybrids (**24**–**27**)(**a**–**c**).

The reason for this is that compounds **33**–**36**(**a**–**c**) have better affinities than the 1,4-linked hybrids **28**–**31**(**a**–**c**) but worse than the 1,2-linked **24**–**27**(**a**–**c**); they have shown similar energy values when interacting with Arg^168^ to **28**–**31**(**a**–**c**) and **24**–**27**(**a**–**c**), and concerning the interaction with Asp^194^, they have similar energy values to **24**–**27**(**a**–**c**) but much better than those of **28**–**31**(**a**–**c**), which do not interact with this amino acid residue.

Therefore, at this point, we decided to synthesize, following the novel convergent pathway that we developed, those 1,3-linked pyrimidine-quinolone hybrids having the naphthalen-2-yl moiety as the hydrophobic tail **35**(**a**–**c**), which has already proven to be the most interesting one towards the inhibition of the *h*LDHA enzyme (Figure 4).

The reason for doing so was to ensure that their biological activity was as predicted. This way, once **35**(**a**–**c**) were synthesized, they were subjected to the determination of their IC_50_ value. The reaction yields, time for the synthesis, and the IC_50_ value of the hybrids **35**(**a**–**c**) are shown in Table 5.

Lower reaction times were required for the synthesis of **35**(**a**–**c**), and yields were better than in the 1,2-linked **26**(**a**–**c**), with the exception of **35b**, and similar to the 1,4-linked **30**(**a**–**c**) due to the meta disposition and thus were not affected by steric hindrance.

From these hybrids, it is important that the chemical shift of the proton located in position 2 of the 3-aminobenzenethiol moiety goes from 6.60 ppm in intermediates **32**(**a**–**c**) to 8.10 ppm in hybrids **35**(**a**–**c**) as a consequence of being linked to the pyrimidine moiety. That proton is coupled with those in positions 4 and 6, with its coupling constant (*J*) being 2.0 Hz in **35a**. In hybrids **35b** and **35c** as well as in intermediates **32**(**a**–**c**), the spectrum is not clear enough in order to differentiate the coupling, being represented as a pseudo-singlet.

From the inhibition assays, it can be said that, as seen in Table 5, compound **35a** has a slight worse inhibitory activity than **26a** (19.6 and 17.8 μM, respectively) and **35b** than **26b** (20.3 and 24.6 μM, respectively), with the difference becoming even larger when comparing **35c** (50.1 μM) to **26c** (27.2 μM).

All of this is in concordance with what was previously predicted: the inhibitory activity of those hybrids having the U-shaped disposition **26**(**a**–**c**) is slightly better than in the case of **35**(**a**–**c**) and both of them drastically better than **30**(**a**–**c**). For an easier interpretation of the results, the different IC_50_ values for the differently linked hybrids are shown in Figure 14.

Results shown in Figure 14 demonstrate that, even though the inhibitory activity of the 1,3-linked pyrimidine-quinolone hybrids **35**(**a**–**c**) is close to those having the 1,2-linked disposition, this type of substitution it is not the best one, which was proven to be that of the 1,2-linked hybrids.

Additionally, it was shown that the effect of having a bulkier group, such as the methoxy one (**8c**), in the quinolone moiety is translated into a slightly lower inhibitory activity.

## 3. Materials and Methods

### 3.1. General

All chemicals and solvents were purchased from Sigma-Aldrich unless stated otherwise. Melting points were collected using a Brastead Electrothermal 9100 melting point apparatus, and the acquired data are uncorrected. IR spectra were recorded on a Fourier Bruker Tensor 27 Spectrophotometer using the ATR dura Sample IR accessory. NMR spectra were recorded in Bruker Avance NEO 400 spectrometer at 400 MHz (^1^H) and 100 MHz (^13^C) at 298 K and 393 K and Bruker Advance 500 spectrometer at 500 MHz (^1^H) and 125 MHz (^13^C) at 298 K and 393 K, using as solvent DMSO-*d*_6_ and as the internal reference tetramethylsilane (0 ppm) or the residual ^1^H/^13^C solvent signals, that is, 2.50/39.52. DEPT-135 and 2D-NMR (HSQC, HMBC, and COSY) experiments were used for the assignment of carbon and hydrogen signals. Chemical shifts (δ) are given in ppm, and coupling constants (*J*) are given in Hz. The following abbreviations are used for multiplicities: s, singlet; d, doublet; t, triplet; q, quartet; m, multiplet; ps, pseudo-singlet; pd, pseudo-doublet; and pt, pseudo-triplet. The mass spectra were recorded on a Thermo model DSQ II spectrometer equipped with a direct inlet probe and operating at 70 eV. HPLC-HRMS data were obtained on an Agilent Technologies Q-TOF 6530B coupled to an HPLC Agilent-1260 Infinity, equipped with a Kinetex C18 column (2.1 mm × 50 mm × 2.6 um) PN 00B-4462-AN using the following HPLC method: flow, 0.4 mL/min; elution gradient, 0–5 min from acetonitrile/water 10% (0.1% formic acid) to acetonitrile 100% (0.1% formic acid); plus 3 additional minutes at that concentration. Ionization method: electrospray ionization; (ESI+) acquisition software: MassHunter LC/MS Data Acquisition 6200 series TOD/6500 series Q-TOF, Version: B.06.01 (Build 6.01.6172 SP1). The single-crystal X-ray data were collected in a Diffractometer Bruker D8 Venture. All the equipment used in the spectroscopic and spectrometric analysis belong to “Centro de Instrumentación Científico y Técnico”, (CICT) in “Universidad de Jaén” (UJA). The reactions were monitored by TLC on a 0.2mm pre-coated aluminum plates of silica gel (Merck 60 F_254_), and spots were visualized by UV irradiation (254nm). All reagents were purchased from commercial sources and used without further purification unless otherwise noted. All starting materials were weighed and handled in air at room temperature. Precursor quinolone derivatives (**8**(**a**–**c**)) [67] and 4-aryl-2-chloropyrimidines (**1**–**4**) [58] were prepared according to reported procedures.

### 3.2. Chemistry

#### 3.2.1. Synthesis of **4-(4-Chlorophenyl)-N-(4-((2-chloroquinolin-3-yl)methoxy)phenyl)pyrimidin-2-amine** (**8**)

3-(Bromomethyl)-2-chloroquinoline (0.30 mmol) was added to a solution of **5** (0.30 mmol) and potassium carbonate (0.60 mmol) in acetonitrile (5mL). The mixture was heated up at reflux within 4 h and 25 min. Once the reaction was completed (TLC monitored), the mixture was cooled at room temperature, and the solid was filtered and washed with cold acetonitrile and water, respectively. No further purification was done. Yellow Solid (64%) M.p. 478–481 K. Rf Hex:AcOEt (6:4): 0.27. ^1^H NMR (400 MHz, DMSO-d_6_) δ 9.60 (s, 1H), 8.58 (d, *J* = 22.7 Hz, 1H), 8.52 (d, *J* = 5.2 Hz, 1H), 8.19–8.14 (m, 2H), 8.09 (d, *J* = 8.0 Hz, 1H), 8.00 (dd, *J* = 8.0, 6.9 Hz, 1H), 7.84 (ddd, *J* = 8.0, 6.9, 1.5 Hz, 1H), 7.74 (pd, *J* = 9.0 Hz, 2H), 7.71–7.67 (m, 1H), 7.63–7.58 (m, 2H), 7.36 (d, *J* = 5.2 Hz, 1H), 7.09 (dd, *J* = 9.0, 2.0 Hz, 2H), and 5.26 (d, *J* = 17.4 Hz, 2H). ^13^C NMR (100 MHz, DMSO-d_6_) δ 162.3, 160.2, 159.2, 152.9, 149.0, 147.2, 146.5, 142.1, 138.2, 137.7, 135.59, 135.57, 134.4, 130.9, 130.7, 128.99, 128.93, 128.6, 128.2, 128.1, 127.7, 127.6, 127.6, 126.9, 126.9, 120.7, 115.0, 107.4, 68.5, and 66.8. IR (ATR, cm^−1^): 3275 (NH), 3195, 3120, 1508, 1425, 1229, 804, and 748. EI MS (70eV): *m*/*z* (%): 472 (M+, 1), 298 (37), 196 (100), 140 (16), and 44 (69). HRMS (ESI-QTOF) M + H calc. for C_26_H_18_Cl_2_N_4_O: 473.0930 found: 473.0927.

#### 3.2.2. Synthesis of **3-((2-Hydroxyphenoxy)methyl)quinolin-2(1*H*)-one** (**9**)

1,2-Dihydroxybenzene (4.30 mmol) was added to a solution of **6a** (0.86 mmol) and potassium carbonate (1.72 mmol) in THF (3mL). The mixture was stirred at room temperature for 13h. After the reaction was completed (TLC monitored), the solvent was removed under vacuum, and water was added, introducing the mixture under ultrasound in order to enable the precipitation. After that, the solid was collected by filtration. The desired product was obtained by further purification with DCM:MeOH, 97:3. White Solid (57%) M.p. 496–499 K. Rf DCM:MeOH, (97:3): 0.28. ^1^H NMR (400 MHz, DMSO-d_6_) δ 12.06 (s, 1H), 9.21 (s, 1H), 8.14 (s, 1H), 7.70 (dd, *J* = 7.2, 1.5 Hz, 1H), 7.52 (ddd, *J* = 8.4, 7.2, 1.5 Hz, 1H), 7.36 (d, *J* = 8.4 Hz, 1H), 7.25–7.18 (m, 1H), 7.01 (dd, *J* = 7.9, 1.2 Hz, 1H), 6.89–6.80 (m, 2H), and 6.74 (ddd, *J* = 7.9, 6.6, 2.5 Hz, 1H), 4.95 (s, 2H). ^13^C NMR (100 MHz, DMSO-d_6_) δ 161.3, 147.5, 146.4, 138.1, 136.9, 130.3, 128.6, 127.9, 122.2, 122.1, 119.3, 119.0, 116.2, 115.4, 115.1, and 66.7. IR (ATR, cm^−1^): (3400–2400, wide NH amide and OH signals), 1654, 994, and 740. EI MS (70eV): *m*/*z* (%): 267 (M+, 7), 158 (100), and 130 (28). HRMS (ESI-QTOF) M + H calc. for C_16_H_13_NO_3_: 299′0582 found: 299′0582.

#### 3.2.3. Synthesis of **3-((4-((4-(4-Chlorophenyl)pyrimidin-2-yl)amino)phenoxy)methyl)quinolin-2(1*H*)-one** (**15**)

Acetic acid (10mL) was added to a solution of **8** (0.14 mmol) in water (4mL). The mixture was heated at reflux within 7 h and 10 min. Once the reaction was completed (TLC monitored), the mixture was cooled at room temperature and introduced overnight in the refrigerator in order to enable the precipitation. The desired product was obtained by filtration and washed with water. Yellow Solid (61%) M.p. 575–578 K. Rf Hex:AcOEt (6:4): 0.08. ^1^H NMR (400 MHz, DMSO-d_6_) δ 11.99 (s, 1H), 9.57 (s, 1H), 8.51 (d, *J* = 5.2 Hz, 1H), 8.16 (d, *J* = 8.3 Hz, 2H), 8.00 (s, 1H), 7.72–7.69 (m, 3H), 7.59 (d, *J* = 8.3 Hz, 2H), 7.49 (pt, *J* = 7.7 Hz, 1H), 7.36–7.32 (m, 2H), 7.18 (pt, *J* = 7.7 Hz, 1H), 7.03 (d, *J* = 9.0 Hz, 2H), and 4.98 (s, 2H). ^13^C NMR (100 MHz, DMSO-d_6_) δ 162.3, 160.9, 160.3, 159.2, 153.2, 138.1, 136.3, 135.6, 134.0, 130.1, 128.93, 128.89, 128.6, 127.9, 122.0, 120.7, 120.6, 118.9, 115.0, 114.7, 107.3, and 64.9. IR (ATR, cm^−1^): (3400–2400, wide NH amide signal), 3304 (NH), 3066, 2831, 1661 (C=O), 1554, 1508, 1424, 1231, and 802. EI MS (70eV): *m*/*z* (%): 454 (M+, 6), 298 (33), 296 (100), and 158 (15). HRMS (ESI-QTOF) M + H calc. for C_26_H_19_ClN_4_O_2_: 455.1269 found: 455.1266.

#### 3.2.4. Synthesis of **3-((2-((4-(4-Chlorophenyl)pyrimidin-2-yl)oxy)phenoxy)methyl)quinolin-2(1*H*)-one** (**19**)

Intermediate **1** (0.023 mmol) was added to a solution of **9** (0.023 mmol), potassium carbonate (0.045mmol), and silver nitrate (0.006 mmol) in dimethyl sulfoxide (0.3mL). The mixture was heated up to 80 °C for 8 h and 30 min. After the reaction was completed (TLC monitored), the mixture was neutralized with acetic acid, and the solid was filtered and washed with water. The desired product was obtained by further purification with Hex:AcOEt (4:6). White Solid (67%) M.p. 496–499 K. Rf Hex:AcOEt (4:6): 0.11. ^1^H NMR (400 MHz, DMSO-d_6_) δ 11.88 (s, 1H), 8.68 (d, *J* = 5.2 Hz, 1H), 8.09 (d, *J* = 8.3 Hz, 2H), 7.79 (d, *J* = 5.2 Hz, 1H), 7.58 (d, *J* = 8.3 Hz, 2H), 7.45–7.37 (m, 1H), 7.35–7.31 (m, 1H), 7.29–7.21 (m, 3H), 7.18 (s, 1H), 7.07 (td, *J* = 7.4, 2.1 Hz, 1H), 6.98–6.96 (m, 2H), and 4.93 (s, 2H). ^13^C NMR (100 MHz, DMSO-d_6_) δ 165.0, 164.4, 160.9, 160.4, 149.9, 142.0, 137.7, 136.4, 134.4, 134.3, 130.0, 129.1, 128.8, 128.5, 127.2, 126.4, 122.6, 121.7, 121.5, 118.4, 115.0, 114.3, 112.0, and 65.0. IR (ATR, cm^−1^): (3200–2600, wide NH amide signal), 3331, 3063, 2923, 2853, 1660 (C=O), 1577, 1501, 1434, 1379, 1262, 1087, 818, and 745. EI MS (70eV): *m*/*z* (%): 455 (M+, 8), 158 (100), and 130 (21). HRMS (ESI-QTOF) M + H calc. for C_26_H_18_ClN_3_O_3_: 456.1109 found: 456.1108.

#### 3.2.5. General Procedure for the Synthesis of **3-(((2′-Aminophenyl)thio)methyl)quinolin-2(1H)-ones (22(a–c)), 3-(((4′-aminophenyl)thio)methyl)quinolin-2(1H)-ones (23(a–c)), and 3-(((3′-aminophenyl)thio)methyl)quinolin-2(1H)-ones (32(a–c))**

Potassium carbonate (1.5 mmol per mmol of (**1**–**4**) when using 4-aminothiphenol and 1 mmol per mmol when using 2-aminothiophenol) was added to a solution of the corresponding aminothiophenol (1.2 mmol per mmol of (**1**–**4**) for 4-aminothiphenol and 1 mmol per mmol for 2-aminothiophenol) in ethanol, and it was stirred 5 min at room temperature under argon atmosphere. After that, the different 3-(bromomethyl)quinolin-2(1*H*)-ones **6**(**a**–**c**), (1 mmol) were added, and the mixture was stirred until the reaction was completed; TLC monitored using Hex:AcOEt (6:4) as eluent. Once the reaction was completed, the solid was filtered and washed with ethanol and water to afford a pure solid.

##### 3-(((2-Aminophenyl)thio)methyl)quinolin-2(1*H*)-one (**22a**)

White Solid (86%) M.p. 493–496 K. Rf Hex:AcOEt (6:4): 0.19. ^1^H NMR (400 MHz, DMSO-d_6_) δ 11.89 (s, 1H), 7.51–7.41 (m, 3H), 7.30 (d, *J* = 7.7 Hz, 1H), 7.14–7.12 (m, 2H), 7.04 (t, *J* = 7.9 Hz, 1H), 6.72 (d, *J* = 7.9 Hz, 1H), 6.44 (t, *J* = 7.9 Hz, 1H), 5.63 (s, 2H), and 3.82 (s, 2H). ^13^C NMR (100 MHz, DMSO-d_6_) δ 161.3, 149.6, 138.2, 137.2, 135.7, 129.84, 129.77, 129.0, 127.5, 121.9, 119.0, 116.3, 115.2, 114.9, 114.4, and 32.9. IR (ATR, cm^−1^): (3500–2400, wide NH amide signal), 3475 and 3373 (NH_2_), 3148, 3008, 2848, 1661 (C=O), 1599, and 744. EI MS (70eV): *m*/*z* (%): 282 (M^+^, 23), 158 (100), and 130 (33). HRMS (ESI-QTOF) M + H calc. for C_16_H_14_N_2_OS: 283.0900, found: 283.0895.

##### 3-(((2-Aminophenyl)thio)methyl)-6-chloroquinolin-2(1*H*)-one (**22b**)

White Solid (68%) M.p. 509–512 K. Rf Hex:AcOEt (1:1): 0.24. ^1^H NMR (400 MHz, DMSO-d_6_) δ 12.01 (s, 1H), 7.58 (s, 1H), 7.48 (d, *J* = 8.5 Hz, 1H), 7.44 (s, 1H), 7.29 (d, *J* = 8.5 Hz, 1H), 7.11 (d, *J* = 7.5 Hz, 1H), 7.04 (t, *J* = 7.5 Hz, 1H), 6.70 (d, *J* = 7.5 Hz, 1H), 6.43 (t, *J* = 7.5 Hz, 1H), 5.52 (s, 2H), and 3.79 (s, 2H). ^13^C NMR (100 MHz, DMSO-d_6_) δ 161.1, 149.9, 136.9, 135.9, 135.9, 130.5, 129.9, 129.7, 126.4, 125.6, 120.2, 116.8, 116.1, 114.8, 114.3, and 33.0. IR (ATR, cm^−1^): (3500–2400, wide NH amide signal), 3430 and 3311 (NH_2_), 3156, 2986, 2819, 1661 (C=O), 818, 748, and 589. EI MS (70eV): *m*/*z* (%): 316 (M^+^, 15), 192 (100), 164 (39), and 80 (41). HRMS (ESI-QTOF) M + H calc. for C_16_H_13_ClN_2_OS: 317.0510, found: 317.0513.

##### 3-(((2-Aminophenyl)thio)methyl)-6-methoxyquinolin-2(1*H*)-one (**22c**)

Yellowish Solid (76%) M.p. 469–472 K. Rf Hex:AcOEt (1:1): 0.11. ^1^H NMR (400 MHz, DMSO-d_6_) δ 11.79 (s, 1H), 7.50 (s, 1H), 7.24 (d, *J* = 9.0 Hz, 1H), 7.16–7.13 (m, 1H), 7.12–7.09 (m, 1H), 7.07–7.01 (m, 2H), 6.72 (d, *J* = 8.0 Hz, 1H), 6.45 (t, *J* = 8.0 Hz, 1H), 5.63 (s, 2H), 3.81 (s, 2H),and 3.74 (s, 3H). ^13^C NMR (100 MHz, DMSO-d_6_) δ 160.9, 154.2, 149.7, 136.9, 135.6, 132.7, 129.7, 129.5, 119.6, 119.0, 116.2, 116.2, 115.2, 114.4, 108.9, 55.4, and 33.1. IR (ATR, cm^−1^): (3400–2200, wide NH amide signal), 3443 and 3334 (NH_2_), 3144, 2928, 1604 (C=O), 756, 596, and 460. EI MS (70eV): *m*/*z* (%): 312 (M^+^, 10), 188 (100), 160 (15), and 117 (12). HRMS (ESI-QTOF) M + H calc. for C_17_H_16_N_2_O_2_S: 313.1005, found: 313.1008.

##### 3-(((4-Aminophenyl)thio)methyl)quinolin-2(1*H*)-one (**23a**)

White Solid (82%) M.p. 488–491 K. Rf Hex:AcOEt (1:1): 0.11. ^1^H NMR (400 MHz, DMSO-d_6_) δ 11.83 (s, 1H), 7.47 (d, *J* = 8.0 Hz, 1H), 7.43 (pt, *J* = 7.7 Hz, 1H), 7.40 (s, 1H), 7.28 (d, *J* = 8.4 Hz, 1H), 7.11 (pt, *J* = 7.7 Hz, 1H), 7.03 (d, *J* = 8.5 Hz, 2H), 6.48 (d, *J* = 8.5 Hz, 2H), 5.35 (s, 2H), and 3.78 (s, 2H). ^13^C NMR (100 MHz, DMSO-d_6_) δ 161.2, 148.5, 138.1, 136.5, 134.5, 129.7, 129.7, 127.4, 121.8, 119.0, 118.8, 114.8, 114.4, and 35.8. IR (ATR, cm^−1^): (3500–2400, wide NH amide signal), 3470 and 3374 (NH_2_), 2991, 1658 (C=O), 758, and 495. EI MS (70eV): *m*/*z* (%): 282(M^+^, 15), 158 (100), and 130 (30). HRMS (ESI-QTOF) M + H calc. for C_16_H_14_N_2_OS: 283.0900, found: 283.0897.

##### 3-(((4-Aminophenyl)thio)methyl)-6-chloroquinolin-2(1*H*)-one (**23b**)

White Solid (76%) M.p. 488–491 K. Rf Hex:AcOEt (4:6): 0.38. ^1^H NMR (400 MHz, DMSO-d_6_) δ 11.96 (s, 1H), 7.61 (ps, 1H), 7.46 (pd, *J* = 8.8 Hz, 1H), 7.39 (s, 1H), 7.28 (d, *J* = 8.8 Hz, 1H), 7.02 (d, *J* = 8.5 Hz, 2H), 6.47 (d, *J* = 8.5 Hz, 2H), 5.32 (s, 2H), and 3.76 (s, 2H). ^13^C NMR (101 MHz, DMSO) δ 160.9, 148.8, 136.8, 135.3, 134.7, 131.2, 129.5, 126.3, 125.6, 120.2, 118.3, 116.7, 114.3, and 35.9. IR (ATR, cm^−1^): (3500–2400, wide NH amide signal), 3444 and 3356 (NH_2_), 3224, 3146 (NH), 2992, 2909, 2829, 2732, 1648 (C=O), 1597, and 589. EI MS (70eV): *m*/*z* (%): 316(M^+^, 27), 192 (77), 124 (100), and 93 (34). HRMS (ESI-QTOF) M + H calc. for C_16_H_13_ClN_2_OS: 317.0510, found: 317.0508.

##### 3-(((4-Aminophenyl)thio)methyl)-6-methoxyquinolin-2(1*H*)-one (**23c**)

White Solid (77%) M.p. 459–461 K. Rf Hex:AcOEt (4:6): 0.11. ^1^H NMR (400 MHz, DMSO-d_6_) δ 11.73 (s, 1H), 7.40 (s, 1H), 7.22 (d, *J* = 8.8 Hz, 1H), 7.08 (dd, *J* = 8.8, 2.9 Hz, 1H), 7.06–6.99 (m, 3H), 6.48 (d, *J* = 8.5 Hz, 2H), 5.31 (s, 2H), 3.78 (s, 2H), and 3.74 (s, 3H). ^13^C NMR (100 MHz, DMSO-d_6_) δ 160.7, 154.1, 148.7, 136.2, 134.4, 132.6, 130.2, 119.6, 118.8, 116.1, 114.3, 108.9, 55.4, and 35.9. IR (ATR, cm^−1^): (3500–2400, wide NH amide signal), 3469 and 3357 (NH_2_), 3147, 2928, 1611 (C=O), 810, and 599. EI MS (70eV): *m*/*z* (%): 312 (M^+^, 15), 188 (100), 160 (20), and 117 (20). HRMS (ESI-QTOF) M + H calc. for C_17_H_16_N_2_O_2_S: 313.1005, found: 313.1006.

##### 3-(((3-Aminophenyl)thio)methyl)quinolin-2(1*H*)-one (**32a**)

White Solid (76%) M.p. 434–437 K. Rf Hex:AcOEt (1:1): 0.07. ^1^H NMR (400 MHz, DMSO-d_6_) δ 11.91 (s, 1H), 7.81 (s, 1H), 7.56 (pd, *J* = 8.0 Hz, 1H), 7.45 (pt, *J* = 7.7 Hz, 1H), 7.30 (pd, *J* = 8.4 Hz, 1H), 7.14 (pt, *J* = 7.7 Hz, 1H), 6.96 (pt, *J* = 8.0 Hz, 1H), 6.61 (ps, 1H), 6.53 (pd, *J* = 8.2 Hz, 1H), 6.42 (pd, *J* = 7.8 Hz, 1H), 5.56 (s, 2H), and 4.01 (s, 2H). ^13^C NMR (100 MHz, DMSO-d_6_) δ 161.3, 148.2, 138.1, 136.9, 136.3, 129.9, 129.5, 129.3, 127.6, 121.9, 119.0, 116.4, 114.9, 114.0, 112.3, and 31.7. IR (ATR, cm^−1^): (3400–2200, wide NH amide signal), 3435 and 3313 (NH_2_), 3157, 3023, 1641 (C=O), and 765. EI MS (70eV): *m*/*z* (%): 282 (M+, 9), 158 (100), 130 (52), and 80 (26). HRMS (ESI-QTOF) M + H calc. for C_16_H_14_N_2_OS: 283.0900 found: 283.0897.

##### 3-(((3-Aminophenyl)thio)methyl)-6-chloroquinolin-2(1*H*)-one (**32b**)

White Solid (88%) M.p. 477–480 K. Rf Hex:AcOEt (6:4): 0.22. ^1^H NMR (400 MHz, DMSO-d_6_) δ 12.03 (s, 1H), 7.79 (s, 1H), 7.70 (s, 1H), 7.49 (d, *J* = 8.8 Hz, 1H), 7.29 (d, *J* = 8.8 Hz, 1H), 6.93 (pt, *J* = 8.0 Hz, 1H), 6.55 (ps, 1H), 6.47 (pd, *J* = 7.8 Hz, 1H), 6.38 (pd, *J* = 7.8 Hz, 1H), 5.18 (s, 2H), and 3.99 (s, 2H). ^13^C NMR (100 MHz, DMSO-d_6_) δ 161.0, 149.2, 136.8, 136.0, 135.7, 130.8, 129.8, 129.5, 126.5, 125.7, 120.2, 116.8, 115.9, 113.6, 112.0, and 31.9. IR (ATR, cm^−1^): (3200–2100, wide NH amide signal), 3468 and 3361 (NH2), 3147, 3055, 2979, 2818, 1654 (C=O), and 770. EI MS (70eV): *m*/*z* (%): 316 (M+, 12), 192 (100), 164 (42), and 80 (50). HRMS (ESI-QTOF) M + H calc. for C_16_H_13_ClN_2_OS: 317.0510 found: 317.0510.

##### 3-(((3-Aminophenyl)thio)methyl)-6-methoxyquinolin-2(1*H*)-one (**32c**)

White Solid (79%) M.p. 448–451 K. Rf Hex:AcOEt (4:6): 0.09. ^1^H NMR (400 MHz, DMSO-d_6_) δ 11.81 (s, 1H), 7.79 (s, 1H), 7.24 (d, *J* = 8.8 Hz, 1H), 7.14–7.07 (m, 2H), 6.98 (pt, *J* = 8.0 Hz, 1H), 6.62 (ps, 1H), 6.56 (pd, *J* = 7.8 Hz, 1H), 6.45 (pd, *J* = 7.8 Hz, 1H), 5.76 (s, 2H), 4.02 (s, 2H), and 3.75 (s, 3H). ^13^C NMR (100 MHz, DMSO-d_6_) δ 160.8, 154.2, 147.7, 136.5, 132.6, 129.6, 129.5, 119.6, 119.2, 116.6, 116.2, 114.0, 112.6, 109.0, 55.4, and 31.7. IR (ATR, cm^−1^): (3300–2200, wide NH amide signal), 3443 and 3347 (NH_2_), 3141, 2931, 2830, 1617 (C=O), and 783. EI MS (70eV): *m*/*z* (%): 312 (M+, 8), 188 (100), 160 (23), 117 (26), and 80 (21). HRMS (ESI-QTOF) M + H calc. for C_17_H_16_N_2_O_2_S: 313.1005 found: 313.1005.

#### 3.2.6. General Procedure for the Synthesis of **3-(((2/4-((4-(Aryl)pyrimidin-2-yl)amino)phenyl)thio)methyl)quinolin-2(1*H*)-ones (24–31)(a–c) and 3-(((3-((4-(naphthalen-2-yl)pyrimidin-2-yl)amino)phenyl)thio)methyl)quinolin-2(1*H*)-ones (35(a–c))**

Intermediates (**22**, **23**, and **31**(**a**–**c**), (1 mmol)) were added to a solution of 4-aryl-2-chloropyrimidine ((**1**–**4**), 1 mmol) in EtOH (3mL per mmol). The mixture was subjected to microwave irradiation at 120 °C until the reaction was completed (TLC monitored using Hex:AcOEt (4:6 or 1:1) as eluent), with a setting of 250 psi and 300 W for maximum pressure and power, respectively. After the reaction was completed, the desired product was obtained by filtration and washed with cold EtOH. No purification was needed, but to ensure maximum purity for biological assays, compounds **28**–**31**(**a**–**c**) were recrystallized from DMF, leaving the recipient open to the air and, if necessary, introduced in the refrigerator. Compounds **24**–**27**(**a**–**c**) were recrystallized (after having been filtrated) from ethanol under MW irradiation (1 min at 120 °C). Compounds **35**(**a**–**c**) were recrystallized from EtOH, leaving the recipient open to the air.

##### 3-(((2-((4-(4-Chlorophenyl)pyrimidin-2-yl)amino)phenyl)thio)methyl)quinolin-2(1*H*)-one (**24a**)

White Solid (86%) M.p. 460–463 K. Rf Hex:AcOEt (6:4): 0.15. ^1^H NMR (400 MHz, DMSO-d_6_) δ 11.32 (s, 1H), 8.46–8.44 (m, 2H), 8.24 (dd, *J* = 8.2, 1.4 Hz, 1H), 8.11–8.02 (m, 2H), 7.57–7.55 (m, 3H), 7.44 (s, 1H), 7.37–7.35 (m, 2H), 7.31 (d, *J* = 8.0 Hz, 1H), 7.30 (d, *J* = 5.2 Hz, 1H), 7.20 (d, *J* = 8.4 Hz, 1H), 7.08–6.96 (m, 2H), and 3.97 (s, 2H). ^13^C NMR (100 MHz, DMSO-d_6_) δ 162.1, 160.3, 159.3, 158.3, 140.3, 137.7, 136.3, 135.1, 134.9, 134.0, 128.9, 128.5, 128.2, 128.1, 128.0, 126.6, 124.1, 122.2, 120.8, 120.2, 118.4, 114.2, 108.0, and 34.5. IR (ATR, cm^−1^): (3600–2400, wide NH amide signal), 3318 (NH), 3161, 1658 (C=O), 1526, 1436, and 753. EI MS (70eV): *m*/*z* (%): 470 (M^+^, 15), 312 (72), 280 (44), 158 (100), and 130 (50). HRMS (ESI-QTOF) M + H calc. for C_26_H_19_ClN_4_O_2_: 471.1041, found: 471.1042.

##### 6-Chloro-3-(((2-((4-(4-chlorophenyl)pyrimidin-2-yl)amino)phenyl)thio)meth-yl)quinolin-2(1*H*)-one (**24b**)

Yellowish Solid (61%). M.p. 509–512 K. Rf Hex:AcOEt (1:1): 0.30. ^1^H NMR (400 MHz, DMSO-d_6_) δ 11.91 (s, 1H), 8.90 (s, 1H), 8.45 (d, *J* = 5.2 Hz, 1H), 8.05 (d, *J* = 8.3 Hz, 2H), 8.01 (pd, *J* = 8.2 Hz, 1H), 7.60–7.55 (m, 3H), 7.41–7.35 (m, 4H), 7.26 (dd, *J* = 8.8, 2.4 Hz, 1H), 7.12 (pt, *J* = 7.5 Hz, 1H), 7.09 (d, *J* = 8.8 Hz, 1H), and 3.89 (s, 2H). ^13^C NMR (100 MHz, DMSO-d_6_) δ 163.2, 160.7, 158.5, 157.5, 139.9, 136.7, 136.1, 135.7, 134.7, 134.2, 130.1, 129.5, 129.5, 129.0, 128.9, 126.2, 125.9, 125.5, 124.1, 122.2, 119.9, 116.6, 108.2, and 34.9. IR (ATR, cm^−1^): (3600–2000, wide NH amide signal), 3336 (NH), 3157, 3056, 2832, 1656 (C=O), 1568, 1513, 1433, and 745. EI MS (70eV): *m*/*z* (%): 504 (M^+^, 7), 312 (60), 280 (82), 192 (41), 164 (37), and 43 (100). HRMS (ESI-QTOF) M + H calc. for C_26_H_18_Cl_2_N_4_OS: 505.0651, found: 505.0650. Crystals suitable for X-ray single-crystal diffraction were obtained from DMSO solution, and the crystal data for **24b** DMSO solvate were deposited at CCDC with reference CCDC 2159307: Chemical formula C_26_H_18_Cl_2_N_4_OS · C_2_H_6_OS, Mr 739,107; Monoclinic, C2/c; 116K, Cell dimensions a, b, c (Å)48.8146 (17), 5.1928 (1), 34.4150 (13) β (°) α, β, γ (º) 90, 127.875 (1), 90. V (Å^3^) 6886.0 (4), Z = 8, F (000) = 2416, Dx (Mg m^−3^) = 1.13, Mo Kα, μ (mm^−1^) = 0.47, Crystal size (mm) = 0.4 × 0.22 × 0.06. Data collection: Diffractometer Bruker D8 Venture (APEX 3), Monochromator multilayer mirror, CCD rotation images, thick slices φ and θ scans, Mo INCOATEC high-brilliance microfocus sealed tube (λ = 0.71073 Å), multiscan absorption correction (SADABS 2016/2), Tmin, Tmax 0.660, 0.746. No. of measured, independent and observed [I > 2σ(I)] reflections 116,172, 7911, 7618, Rint = 0. 056, (sin θ/λ)max (Å^−1^) 0.650, θ values (°): θmax = 28.3, θmin = 2.1; Range h = −62→62, k = −6→6, l = −44→44, Refinement on F^2^:R[F^2^ > 2σ(F^2^)] = 0. 096, wR(F^2^) = 0. 128, S=1.121. No. of reflections 8554, No. of parameters 345, No. of restraints 277. Weighting scheme: w = 1/σ^2^(Fo^2^) + (0.0432P)^2^ + 24.4231P where P = (Fo^2^ + 2Fc^2^)/3. (∆/σ) < 0.001, Δρmax, Δρmin (e Å^−3^) 0.783, −0.43. Several molecules of disorder DMSO were found in the difference map and the above data resulted from application of Squeeze (Version = 260918).

##### 3-(((2-((4-(4-Chlorophenyl)pyrimidin-2-yl)amino)phenyl)thio)methyl)-6-methoxyquinolin-2(1*H*)-one (**24c**)

Yellowish Solid (71%) M.p. 401–404 K. Rf Hex:AcOEt (1:1): 0.14. ^1^H NMR (400 MHz, DMSO-d_6_) δ 11.71 (s, 1H), 8.82 (s, 1H), 8.44 (d, *J* = 5.2 Hz, 1H), 8.06 (d, *J* = 8.3 Hz, 2H), 8.04 (pd, *J* = 8.2 Hz, 1H), 7.60–7.54 (m, 3H), 7.39–7.34 (m, 3H), 7.11 (ptd, *J* = 7.5, 1.4 Hz, 1H), 7.07 (d, *J* = 8.8 Hz, 1H), 6.94 (dd, *J* = 8.8, 2.4 Hz, 1H), 6.82 (d, *J* = 2.4 Hz, 1H), 3.91 (s, 2H), and 3.62 (s, 3H). ^13^C NMR (100 MHz, DMSO-d_6_) δ 162.7, 160.6, 159.1, 158.2, 154.0, 139.9, 136.6, 135.9, 135.0, 133.7, 132.6, 129.2, 128.9, 128.8, 128.6, 126.3, 123.9, 122.1, 119.4, 118.8, 116.1, 108.7, 108.2, 55.2, and 34.8. IR (ATR, cm^−1^): (3600–2200, wide NH amide signal), 3331 (NH), 3060, 2826, 1659 (C=O), 1568, 1433, and 744. EI MS (70eV): *m*/*z* (%): 500 (M^+^, 5), 312 (43), 280 (100), and 188 (94). HRMS (ESI-QTOF) M + H calc. for C_27_H_21_ClN_4_O_2_S: 501.1147, found: 501.1138.

##### 3-(((2-((4-(4-(Trifluoromethyl)phenyl)pyrimidin-2-yl)amino)phenyl)thio)methyl)quinolin-2(1*H*)-one (**25a**)

Pale-yellow Solid (41%) M.p. 478–482 K. Rf Hex:AcOEt (6:4): 0.12. ^1^H NMR (400 MHz, DMSO-d_6_) δ 11.81 (s, 1H), 8.75 (s, 1H), 8.48 (d, *J* = 5.2 Hz, 1H), 8.24 (d, *J* = 8.1 Hz, 2H), 8.05 (pd, *J* = 8.2 Hz, 1H), 7.87 (d, *J* = 8.1 Hz, 2H), 7.55 (pd, *J* = 7.8 Hz, 1H), 7.41 (s, 1H), 7.40 (d, *J* = 5.2 Hz, 1H), 7.35 (pt, *J* = 7.5 Hz, 1H), 7.31–7.29 (m, 2H), 7.13 (d, *J* = 8.4 Hz, 1H), 7.09 (pt, *J* = 7.5 Hz, 1H), 6.97 (pt, *J* = 7.7 Hz, 1H), and 3.93 (s, 2H). ^13^C NMR (100 MHz, DMSO-d_6_) δ 162.1, 161.1, 159.8, 159.2, 140.3, 140.1, 138.1, 137.0, 133.7, 130.8, 130.5, 129.7, 128.7, 128.6, 127.7, 127.3, 126.2, 125.7, 123.7, 122.0, 121.6, 118.8, 114.8, 108.9, and 34.6. IR (ATR, cm^−1^): (3700–2100, wide NH amide signal), 3333 (NH), 3060, 2998, 1658 (C=O), 1568, 1520, 1433, and 743. EI MS (70eV): *m*/*z* (%): 504 (M^+^, 6), 346 (40), 314 (25), 158 (100), and 130 (50). HRMS (ESI-QTOF) M + H calc. for C_27_H_19_F_3_N_4_OS: 505.1304, found: 505.1303.

##### 6-Chloro-3-(((2-((4-(4-(trifluoromethyl)phenyl)pyrimidin-2-yl)amino)phenyl) thio)methyl)quinolin-2(1*H*)-one (**25b**)

Yellow Solid (50%) M.p. 509–513 K. Rf Hex:AcOEt (1:1): 0.17. ^1^H NMR (400 MHz, DMSO-d_6_) δ 11.87 (s, 1H), 8.63 (s, 1H), 8.46 (d, *J* = 5.2 Hz, 1H), 8.21 (d, *J* = 8.1 Hz, 2H), 8.07 (pd, *J* = 8.2 Hz, 1H), 7.86 (d, *J* = 8.1 Hz, 2H), 7.62–7.57 (m, 1H), 7.39–7.37 (m, 2H), 7.31 (d, *J* = 2.4 Hz, 1H), 7.27 (s, 1H), 7.22 (dd, *J* = 8.8, 2.4 Hz, 1H), 7.10 (pt, *J* = 7.5 Hz, 1H), 7.04 (d, *J* = 8.8 Hz, 1H), and 3.87 (s, 2H). ^13^C NMR (100 MHz, DMSO-d_6_) δ 161.8, 160.7, 159.6, 159.4, 140.7, 140.2, 136.7, 135.5, 134.5, 130.7, 130.4, 130.1, 129.4, 129.0, 127.6, 126.1, 125.7, 125.5, 125.2, 123.5, 121.6, 119.8, 116.6, 108.9, and 35.2. IR (ATR, cm^−1^): (3200–2400, wide NH amide signal), 3325 (NH), 3160, 2991, 2823, 1659 (C=O), 1523, and 807. EI MS (70eV): *m*/*z* (%): 538 (M^+^, 9), 346 (100), 314 (76), 192 (96), and 164 (73). HRMS (ESI-QTOF) M + H calc. for C_27_H_18_ClF_3_N_4_OS: 539.0915, found: 539.0906.

##### 6-Methoxy-3-(((2-((4-(4-(trifluoromethyl)phenyl)pyrimidin-2-yl)amino)phenyl)thio)methyl)quinolin-2(1*H*)-one (**25c**)

White Solid (57%) M.p. 486–489 K. Rf Hex:AcOEt (1:1): 0.07. ^1^H NMR (400 MHz, DMSO-d_6_) δ 11.69 (s, 1H), 8.69 (s, 1H), 8.47 (d, *J* = 5.2 Hz, 1H), 8.23 (d, *J* = 8.1 Hz, 2H), 8.07 (pd, *J* = 8.2 Hz 1H), 7.86 (d, *J* = 8.1 Hz, 2H), 7.57 (pd, *J* = 7.8 Hz, 1H), 7.38 (d, *J* = 5.2 Hz, 1H), 7.35–7.33 (m, 2H), 7.10 (pt, *J* = 7.5 Hz, 1H), 7.04 (d, *J* = 8.8 Hz, 1H), 6.91 (dd, *J* = 8.8, 2.4 Hz, 1H), 6.80 (d, *J* = 2.4 Hz, 1H), 3.90 (s, 2H), and 3.58 (s, 3H). ^13^C NMR (100 MHz, DMSO-d_6_) δ 161.8, 160.6, 159.8, 159.4, 153.9, 140.3, 140.3, 136.5, 133.9, 132.6, 130.7, 130.4, 129.2, 128.7, 127.6, 125.9, 125.7, 123.6, 121.9, 119.3, 118.7, 116.0, 108.9, 108.7, 55.1, and 35.0. IR (ATR, cm^−1^): (3400–2400, wide NH amide signal), 3337 (NH), 3157, 3055, 3000, 2829, 1660 (C=O), 1514, and 808. EI MS (70eV): *m*/*z* (%): 534 (M^+^, 11), 346 (39), 314 (27), 188 (100), 160 (34), and 117 (32). HRMS (ESI-QTOF) M + H calc. for C_28_H_21_F_3_N_4_O_2_S: 535.1410, found: 535.1409.

##### 3-(((2-((4-(Naphthalen-2-yl)pyrimidin-2-yl)amino)phenyl)thio)methyl)quinolin-2(1*H*)-one (**26a**)

White Solid (65%) M.p. 484–487 K. Rf Hex:AcOEt (1:1): 0.14. ^1^H NMR (400 MHz, DMSO-d_6_) δ 11.84 (s, 1H), 8.73 (s, 1H), 8.68 (s, 1H), 8.47 (d, *J* = 5.2 Hz, 1H), 8.21–8.13 (m, 2H), 8.09–8.02 (m, 2H), 7.98 (pd, *J* = 7.8 Hz 1H), 7.61–7.59 (m, 2H), 7.54 (pd, *J* = 7.8 Hz, 1H), 7.50 (d, *J* = 5.2, 1H), 7.44 (s, 1H), 7.39 (pt, *J* = 7.5 Hz, 1H), 7.34–7.29 (m, 2H), 7.16 (d, *J* = 8.4 Hz, 1H), 7.08 (pt, *J* = 7.5 Hz, 1H), 6.96 (pt, *J* = 7.7 Hz, 1H), and 3.95 (s, 2H). ^13^C NMR (100 MHz, DMSO-d_6_) δ 163.5, 161.1, 160.0, 158.9, 140.4, 138.1, 137.1, 134.1, 133.9, 133.8, 132.7, 129.7, 128.9, 128.7, 128.6, 128.4, 127.6, 127.5, 127.4, 127.1, 126.7, 125.8, 124.0, 123.4, 121.8, 121.7, 118.8, 114.8, 108.7, and 34.5. IR (ATR, cm^−1^): (3400–2200, wide NH amide signal), 3336 (NH), 3158, 3007, 2850, 1661 (C=O), 1570, 1523, 1427, and 744. EI MS (70eV): *m*/*z* (%): 486 (M^+^, 13), 328 (84), 296 (100), and 130 (40). HRMS (ESI-QTOF) M + H calc. for C_30_H_22_N_4_OS: 487.1587, found: 487.1589.

##### 6-Chloro-3-(((2-((4-(naphthalen-2-yl)pyrimidin-2-yl)amino)phenyl)thio)me-thyl)quinolin-2(1*H*)-one (**26b**)

Yellow Solid (74%) M.p. 501–504 K. Rf Hex:AcOEt (1:1): 0.22. ^1^H NMR (400 MHz, DMSO-d_6_) δ 11.92 (s, 1H), 8.68 (s, 1H), 8.66 (s, 1H), 8.45 (d, *J* = 5.2 Hz, 1H), 8.20–8.14 (m, 2H), 8.06–8.04 (m, 2H), 8.01–7.96 (m, 1H), 7.63–7.55 (m, 3H), 7.49 (d, *J* = 5.2 Hz, 1H), 7.42 (pt, *J* = 7.5 Hz, 1H), 7.37–7.35 (m, 2H), 7.27 (dd, *J* = 8.8, 2.4 Hz, 1H), 7.11–7.09 (m, 2H), and 3.91 (s, 2H). ^13^C NMR (100 MHz, DMSO-d_6_) δ 163.5, 160.8, 159.6, 158.6, 140.6, 136.7, 135.7, 134.3, 134.1, 133.7, 132.7, 130.2, 129.5, 129.0, 128.4, 127.6, 127.5, 127.1, 126.6, 126.2, 125.5, 125.2, 123.9, 123.4, 121.7, 119.9, 116.6, 108.8, and 35.0. IR (ATR, cm^−1^): (3400–2400, wide NH amide signal), 3325 (NH), 3146, 2984, 2823, 1659 (C=O), 1518, and 809. EI MS (70eV): *m*/*z* (%): 520 (M^+^, 6), 328 (65), and 296 (100). HRMS (ESI-QTOF) M + H calc. for C_30_H_21_ClN_4_OS: 521.1197, found: 521.1179.

##### 6-Methoxy-3-(((2-((4-(naphthalen-2-yl)pyrimidin-2-yl)amino)phenyl)thio)methyl)quinolin-2(1*H*)-one (**26c**)

Yellowish Solid (59%) M.p. 412–415 K. Rf Hex:AcOEt (1:1): 0.04. ^1^H NMR (400 MHz, DMSO-d_6_) δ 11.73 (s, 1H), 8.71 (s, 1H), 8.67 (s, 1H), 8.47 (d, *J* = 5.2 Hz, 1H), 8.21–8.15 (m, 2H), 8.08–8.01 (m, 2H), 7.98 (pd, *J* = 7.8 Hz, 1H), 7.64–7.58 (m, 2H), 7.56 (dd, *J* = 7.8, 1.4 Hz, 1H), 7.49 (d, *J* = 5.2 Hz, 1H), 7.43–7.38 (m, 2H), 7.13–7.06 (m, 2H), 6.94 (dd, *J* = 8.8, 2.4 Hz, 1H), 6.83 (d, *J* = 2.4 Hz, 1H), 3.94 (s, 2H), and 3.55 (s, 3H). ^13^C NMR (100 MHz, DMSO-d_6_) δ 163.5, 160.6, 159.8, 158.8, 154.0, 140.4, 136.6, 134.1, 133.8, 132.7, 132.6, 129.2, 128.9, 128.6, 128.4, 127.6, 127.5, 127.1, 126.7, 125.9, 123.9, 123.4, 121.8, 119.4, 118.8, 116.1, 108.7, 108.7, 55.1, and 34.9. IR (ATR, cm^−1^): (3600–2200, wide NH amide signal), 3333 (NH), 3155, 3054, 2996, 2830, 1657 (C=O), 1566, 1504, 1432, and 806. EI MS (70eV): *m*/*z* (%): 516 (M^+^, 23), 328 (86), 296 (100), 188 (92), and 117 (44). HRMS (ESI-QTOF) M + H calc. for C_31_H_24_N_4_O_2_S: 517.1693, found: 517.1692.

##### 3-(((2-((4-(Benzo[d][1,3]dioxol-5-yl)pyrimidin-2-yl)amino)phenyl)thio)methyl)quinolin-2(1*H*)-one (**27a**)

Yellow Solid (68%) M.p. 508–511 K. Rf Hex:AcOEt (4:6): 0.36. ^1^H NMR (400 MHz, DMSO-d_6_) δ 11.83 (s, 1H), 9.11 (s, 1H), 8.40 (d, *J* = 5.2 Hz, 1H), 7.96 (dd, *J* = 8.2, 1.4 Hz, 1H), 7.70 (dd, *J* = 8.1, 1.8 Hz, 1H), 7.61 (d, *J* = 1.8 Hz, 1H), 7.54 (dd, *J* = 7.8, 1.4 Hz, 1H), 7.48 (s, 1H), 7.40–7.30 (m, 4H), 7.17 (d, *J* = 8.4 Hz, 1H), 7.13 (ptd, *J* = 7.5, 1.4 Hz, 1H), 7.06 (d, *J* = 8.1 Hz, 1H), 7.01 (pt, *J* = 7.7, 1H), 6.14 (s, 2H), and 3.95 (s, 2H). ^13^C NMR (101 MHz, DMSO) δ 164.4, 161.1, 157.9, 155.9, 150.3, 148.0, 139.1, 138.1, 137.1, 133.3, 129.9, 129.8, 128.7, 128.4, 127.4, 127.2, 124.4, 122.9, 122.6, 121.7, 118.8, 114.8, 108.6, 107.6, 107.0, 101.9, and 34.1. IR (ATR, cm^−1^): (3600–2200, wide NH amide signal), 3330 (NH), 3059, 2849, 1656 (C=O), 1569, 1518, 1436, and 748. EI MS (70eV): *m*/*z* (%): 480 (M^+^, 15), 322 894), 290 (77), 158 (100), and 130 (85). HRMS (ESI-QTOF) M + H calc. for C_27_H_20_N_4_O_3_S: 481.1329, found: 481.1327.

##### 3-(((2-((4-(Benzo[d][1,3]dioxol-5-yl)pyrimidin-2-yl)amino)phenyl)thio)methyl)-6-chloroquinolin-2(1*H*)-one (**27b**)

Yellowish Solid (69%) M.p. 485–488 K. Rf Hex:AcOEt (1:1): 0.11. ^1^H NMR (400 MHz, DMSO-d_6_) δ 11.90 (s, 1H), 8.71 (s, 1H), 8.35 (d, *J* = 5.2 Hz, 1H), 8.03 (pd, *J* = 8.2 Hz, 1H), 7.64 (dd, *J* = 8.1, 1.8 Hz, 1H), 7.57 (d, *J* = 1.8 Hz, 1H), 7.56 (pd, *J* = 7.8 Hz, 1H), 7.37 (ptd, *J* = 7.5, 1.4 Hz, 1H), 7.34 (d, *J* = 2.4 Hz, 1H), 7.31 (s, 1H), 7.30–7.25 (m, 2H), 7.10–7.08 (m, 2H), 7.03 (d, *J* = 8.1 Hz, 1H), 6.12 (s, 2H), and 3.88 (s, 2H). ^13^C NMR (100 MHz, DMSO-d_6_) δ 163.4, 160.8, 158.7, 157.4, 149.9, 148.0, 140.3, 136.7, 135.7, 134.3, 130.2, 130.1, 129.5, 128.9, 126.2, 125.6, 125.5, 123.7, 122.1, 121.9, 119.9, 116.6, 108.5, 107.8, 106.8, 101.7, and 34.9. IR (ATR, cm^−1^): (3400–2400, wide NH amide signal), 3330 (NH), 3155, 2989, 2888, 1662 (C=O), 1503, and 801. EI MS (70eV): *m*/*z* (%): 514 (M^+^, 20), 322 (100), 290 (80), 192 (32), and 164 (32). HRMS (ESI-QTOF) M + H calc. for C_27_H_19_ClN_4_O_3_S: 515.0939, found: 515.0925.

##### 3-(((2-((4-(Benzo[d][1,3]dioxol-5-yl)pyrimidin-2-yl)amino)phenyl)thio)methyl)-6-methoxyquinolin-2(1*H*)-one (**27c**)

Yellow Solid (58%) M.p. 515–518 K. Rf Hex:AcOEt (1:1): 0.13. ^1^H NMR (400 MHz, DMSO-d_6_) δ 11.72 (s, 1H), 9.14 (s, 1H), 8.40 (d, *J* = 5.2 Hz, 1H), 7.95 (pd, *J* = 8.2 Hz, 1H), 7.69 (pd, *J* = 8.1 Hz, 1H), 7.61–7.54 (m, 2H), 7.43–7.34 (m, 3H), 7.15 (pt, *J* = 7.5 Hz, 1H), 7.08–7.06 (m, 2H), 6.94 (dd, *J* = 8.8, 2.4 Hz, 1H), 6.85 (d, *J* = 2.4 Hz, 1H), 6.13 (s, 2H), 3.94 (s, 2H), and 3.63 (s, 3H). ^13^C NMR (100 MHz, DMSO-d_6_) δ 164.6, 160.6, 157.4, 155.3, 154.0, 150.4, 148.0, 139.0, 136.7, 133.4, 132.6, 129.7, 129.1, 128.4, 127.2, 124.6, 122.9, 122.7, 119.4, 118.9, 116.1, 108.7, 108.6, 107.6, 107.0, 101.9, 55.2, and 34.5. IR (ATR, cm^−1^): (3600–2200, wide NH amide signal), 3332 (NH), 3157, 3062, 2903, 1578 (C=O), 1501, 1441, and 793. EI MS (70eV): *m*/*z* (%): 510 (M^+^, 26), 322 (100), 290 (44), and 188 (92). HRMS (ESI-QTOF) M + H calc. for C_28_H_22_N_4_O_4_S: 511.1435, found: 511.1424.

##### 3-(((4-((4-(4-Chlorophenyl)pyrimidin-2-yl)amino)phenyl)thio)methyl)quinolin-2(1*H*)-one (**28a**)

Pale-yellow Solid (93%) M.p. 546–549 K. Rf Hex:AcOEt (1:1): 0.22. ^1^H NMR (400 MHz, DMSO-d_6_) δ 11.88 (s, 1H), 9.81 (s, 1H), 8.56 (d, *J* = 5.2 Hz, 1H), 8.16 (d, *J* = 8.3 Hz, 2H), 7.78 (d, *J* = 8.5 Hz, 2H), 7.65 (s, 1H), 7.60 (d, *J* = 8.3 Hz, 2H), 7.52 (d, *J* = 8.0 Hz, 1H), 7.45 (pt, *J* = 7.7 Hz, 1H), 7.42 (d, *J* = 5.2 Hz, 2H), 7.34 (d, *J* = 8.5 Hz, 2H), 7.28 (d, *J* = 8.4 Hz, 1H), 7.10 (pt, *J* = 7.7 Hz, 1H), and 3.99 (s, 2H). ^13^C NMR (100 MHz, DMSO-d_6_) δ 162.4, 161.2, 159.9, 159.3, 139.6, 138.1, 136.8, 135.7, 135.4, 131.5, 129.8, 129.5, 129.0, 128.7, 127.5, 126.7, 121.8, 119.3, 119.0, 114.8, 108.0, and 33.9. IR (ATR, cm^−1^): (3400–2400, wide NH amide signal), 3260 (NH), 3172, 2999, 2855, 2361, 1664 (C=O), 1569, 1415, and 800. EI MS (70eV): *m*/*z* (%): 470 (M^+^, 10), 312 (35), 158 (100), and 130 (27). HRMS (ESI-QTOF) M + H calc. for C_26_H_19_ClN_4_OS: 471.1041, found: 471.1043.

##### 6-Chloro-3-(((4-((4-(4-chlorophenyl)pyrimidin-2-yl)amino)phenyl)thio)meth-yl)quinolin-2(1*H*)-one (**28b**)

Yellow Solid (86%) M.p. 541–544 K. Rf Hex:AcOEt (4:6): 0.57. ^1^H NMR (400 MHz, DMSO-d_6_) δ 12.01 (s, 1H), 9.82 (s, 1H), 8.56 (d, *J* = 5.2 Hz, 1H), 8.16 (d, *J* = 8.3 Hz, 2H), 7.78 (d, *J* = 8.5 Hz, 2H), 7.66 (ps, 1H), 7.63 (s, 1H), 7.60 (d, *J* = 8.3 Hz, 2H), 7.47 (dd, *J* = 8.8, 2.4 Hz, 1H), 7.42 (d, *J* = 5.2 Hz, 1H), 7.33 (d, *J* = 8.5 Hz, 2H), 7.28 (d, *J* = 8.8 Hz, 1H), and 3.97 (s, 2H). ^13^C NMR (100 MHz, DMSO-d_6_) δ 162.4, 160.9, 159.9, 159.3, 139.7, 136.8, 135.7, 135.7, 135.4, 131.7, 130.9, 129.7, 129.0, 128.7, 126.5, 126.5, 125.7, 120.2, 119.3, 116.7, 108.1, and 34.0. IR (ATR, cm^−1^): (3300–2400, wide NH amide signal), 3259 (NH), 3174, 3001, 2882, 1669 (C=O), 1570, 1421, and 796. EI MS (70eV): *m*/*z* (%): 504 (M^+^, 11), 471 (16), 312 (100), 192 (39), 164 (23), and 101 (18). HRMS (ESI-QTOF) M + H calc. for C_26_H_18_Cl_2_N_4_OS: 505.0651, found: 505.0656.

##### 3-(((4-((4-(4-Chlorophenyl)pyrimidin-2-yl)amino)phenyl)thio)methyl)-6-methoxyquinolin-2(1*H*)-one (**28c**)

Yellow Solid (90%) M.p. 524–527 K. Rf Hex:AcOEt (4:6): 0.14. ^1^H NMR (400 MHz, DMSO-d_6_) δ 11.79 (s, 1H), 9.81 (s, 1H), 8.55 (d, *J* = 5.2 Hz, 1H), 8.15 (d, *J* = 8.3 Hz, 2H), 7.78 (d, *J* = 8.5 Hz, 2H), 7.62 (s, 1H), 7.58 (d, *J* = 8.3 Hz, 2H), 7.41 (d, *J* = 5.2 Hz, 1H), 7.34 (d, *J* = 8.5 Hz, 2H), 7.23 (d, *J* = 8.8 Hz, 1H), 7.13–7.04 (m, 2H), 4.00 (s, 2H), and 3.70 (s, 3H). ^13^C NMR (100 MHz, DMSO-d_6_) δ 162.4, 160.8, 160.0, 159.2, 154.1, 139.5, 136.5, 135.7, 135.4, 132.6, 131.3, 129.9, 129.0, 128.7, 126.9, 119.6, 119.4, 119.0, 116.1, 108.9, 108.0, 55.3, and 33.9. IR (ATR, cm^−1^): (3400–2400, wide NH amide signal), 3266 (NH), 3169, 3000, 2362, 1666 (C=O), 1568, 1419, and 794. EI MS (70eV): *m*/*z* (%): 500 (M^+^, 5), 312 (100), 280 (34), 188 (92), 140 (23), and 117 (21). HRMS (ESI-QTOF) M + H calc. for C_27_H_21_ClN_4_O_2_S: 501.1147, found: 501.1149.

##### 3-(((4-((4-(4-(Trifluoromethyl)phenyl)pyrimidin-2-yl)amino)phenyl)thio)methyl)quinolin-2(1*H*)-one (**29a**)

Yellow Solid (93%) M.p. 534–537 K. Rf Hex:AcOEt (1:1): 0.26. ^1^H NMR (400 MHz, DMSO-d_6_) δ 11.89 (s, 1H), 9.88 (s, 1H), 8.61 (d, *J* = 5.2 Hz, 1H), 8.33 (d, *J* = 8.1 Hz, 2H), 7.89 (d, *J* = 8.1 Hz, 2H), 7.79 (d, *J* = 8.5 Hz, 2H), 7.65 (s, 1H), 7.51 (d, *J* = 8.0 Hz, 1H), 7.48 (d, *J* = 5.2 Hz, 1H), 7.43 (pt, *J* = 7.7 Hz, 1H), 7.35 (d, *J* = 8.5 Hz, 2H), 7.28 (d, *J* = 8.4 Hz, 1H), 7.09 (pt, *J* = 7.7 Hz, 1H), and 4.00 (s, 2H). ^13^C NMR (100 MHz, DMSO-d_6_) δ 162.1, 161.2, 160.0, 159.5, 140.5, 139.5, 138.1, 136.9, 131.5, 130.8, 130.5, 129.8, 129.5, 127.7, 127.5, 126.9, 125.8, 125.8, 121.8, 119.4, 119.0, 114.9, 108.7, and 33.9. IR (ATR, cm^−1^): (3400–2400, wide NH amide signal), 3257 (NH), 3170, 3000, 2362, 1659 (C=O), 1570, 1417, and 799. EI MS (70eV): *m*/*z* (%): 504 (M^+^, 10), 346 (21), 158 (100), and 130 (22). HRMS (ESI-QTOF) M + H calc. for C_27_H_19_F_3_N_4_OS: 505.1304, found: 505.1305.

##### 6-Chloro-3-(((4-((4-(4-(trifluoromethyl)phenyl)pyrimidin-2-yl)amino)phenyl) thio)methyl)quinolin-2(1*H*)-one (**29b**)

Yellow Solid (88%) M.p. 547–550 K. Rf Hex:AcOEt (1:1): 0.15. ^1^H NMR (400 MHz, DMSO-d_6_) δ 12.01 (s, 1H), 9.89 (s, 1H), 8.61 (d, *J* = 5.2 Hz, 1H), 8.32 (d, *J* = 8.1 Hz, 2H), 7.88 (d, *J* = 8.1 Hz, 2H), 7.79 (d, *J* = 8.5 Hz, 2H), 7.65–7.60 (m, 2H), 7.48 (d, *J* = 5.2 Hz, 1H), 7.45 (dd, *J* = 8.8, 2.4 Hz, 1H), 7.34 (d, *J* = 8.5 Hz, 2H), 7.28 (d, *J* = 8.8 Hz, 1H), and 3.98 (s, 2H). ^13^C NMR (100 MHz, DMSO-d_6_) δ 162.1, 161.0, 160.0, 159.5, 140.5, 139.6, 136.8, 135.7, 131.7, 130.9, 130.8, 130.5, 129.7, 127.7, 126.6, 126.4, 125.8, 125.7, 120.1, 119.4, 116.7, 108.7, and 34.0. IR (ATR, cm^−1^): (3300–2600, wide NH amide signal), 3258 (NH), 3170, 2999, 2919, 1667 (C=O), 1423, 1323, and 797. EI MS (70eV): *m*/*z* (%): 538 (M^+^, 19), 346 (100), 192 (61), 164 (25), and 151 (13). HRMS (ESI-QTOF) M + H calc. for C_27_H_18_ClF_3_N_4_OS: 539.0915, found: 539.0917.

##### 6-Methoxy-3-(((4-((4-(4-(trifluoromethyl)phenyl)pyrimidin-2-yl)amino)phenyl)thio)methyl)quinolin-2(1*H*)-one (**29c**)

Yellow Solid (90%) M.p. 520–523 K. Rf Hex:AcOEt (4:6): 0.14. ^1^H NMR (400 MHz, DMSO-d_6_) δ 11.78 (s, 1H), 9.89 (s, 1H), 8.61 (d, *J* = 5.2 Hz, 1H), 8.33 (d, *J* = 8.1 Hz, 2H), 7.89 (d, *J* = 8.1 Hz, 2H), 7.79 (d, *J* = 8.5 Hz, 2H), 7.63 (s, 1H), 7.48 (d, *J* = 5.2 Hz, 1H), 7.34 (d, *J* = 8.5 Hz, 2H), 7.22 (d, *J* = 8.8 Hz, 1H), 7.10–7.06 (m, 2H), 4.00 (s, 2H), and 3.69 (s, 3H). ^13^C NMR (100 MHz, DMSO-d_6_) δ 162.1, 160.7, 160.0, 159.6, 154.1, 140.5, 139.4, 136.5, 132.6, 131.2, 130.8, 130.5, 129.8, 127.7, 127.0, 125.8, 119.6, 119.5, 119.0, 116.1, 108.9, 108.6, 55.3, and 33.9. IR (ATR, cm^−1^): (3400–2600, wide NH amide signal), 3274 (NH), 3172, 2998, 2833, 2362, 1664 (C=O), 1570, 1415, and 810. EI MS (70eV): *m*/*z* (%): 534 (M^+^, 9), 501 (12), 346 (14), 188 (100), 160 (15), and 117 (14). HRMS (ESI-QTOF) M + H calc. for C_28_H_21_F_3_N_4_O_2_S: 535.1410, found: 535.1416.

##### 3-(((4-((4-(Naphthalen-2-yl)pyrimidin-2-yl)amino)phenyl)thio)methyl)quinolin-2(1*H*)-one (**30a**)

Yellow Solid (86%) M.p. 554–557 K. Rf Hex:AcOEt (6:4): 0.17. ^1^H NMR (400 MHz, DMSO-d_6_) δ 11.90 (s, 1H), 9.84 (s, 1H), 8.76 (s, 1H), 8.60 (d, *J* = 5.2 Hz, 1H), 8.27 (d, *J* = 8.5 Hz, 1H), 8.07 (d, *J* = 8.5 Hz, 2H), 7.99 (pd, *J* = 7.8 Hz, 1H), 7.85 (d, *J* = 8.5 Hz, 2H), 7.67 (s, 1H), 7.64–7.56 (m, 3H), 7.53 (d, *J* = 8.0 Hz, 1H), 7.44 (pt, *J* = 7.7 Hz, 1H), 7.38 (d, *J* = 8.5 Hz, 2H), 7.29 (d, *J* = 8.4 Hz, 1H), 7.10 (pt, *J* = 7.7 Hz, 1H), and 4.01 (s, 2H). ^13^C NMR (100 MHz, DMSO-d_6_) δ 163.6, 161.2, 160.0, 159.0, 139.7, 138.1, 136.8, 134.1, 134.0, 132.7, 131.5, 129.9, 129.5, 128.9, 128.5, 127.7, 127.5, 127.1, 126.8, 126.6, 123.9, 121.8, 119.3, 119.0, 114.9, 108.4, and 33.9. IR (ATR, cm^−1^): (3400–2400, wide NH amide signal), 3266 (NH), 3170, 3001, 2362, 1652 (C=O), 1570, 1415, and 805. EI MS (70eV): *m*/*z* (%): 486 (M^+^, 29), 453 (40), 328 (100), 158 (95), and 152 (38). HRMS (ESI-QTOF) M + H calc. for C_30_H_22_N_4_OS: 487.1587, found: 487.1591.

##### 6-Chloro-3-(((4-((4-(naphthalen-2-yl)pyrimidin-2-yl)amino)phenyl)thio)methyl)quinolin-2(1*H*)-one (**30b**)

Yellow Solid (89%) M.p. 538–541 K. Rf Hex:AcOEt (4:6): 0.44. ^1^H NMR (500 MHz, DMSO-d_6_) δ 12.01 (s, 1H), 9.84 (s, 1H), 8.76 (s, 1H), 8.60 (d, *J* = 5.2 Hz, 1H), 8.27 (dd, *J* = 8.5, 1.8 Hz, 1H), 8.10–8.03 (m, 2H), 8.01–7.97 (m, 1H), 7.86 (d, *J* = 8.5 Hz, 2H), 7.67–7.64 (m, 2H), 7.63–7.59 (m, 2H), 7.57 (d, *J* = 5.2 Hz, 1H), 7.46 (dd, *J* = 8.8, 2.4 Hz, 1H), 7.38 (d, *J* = 8.5 Hz, 2H), 7.30 (d, *J* = 8.8 Hz, 1H), and 4.00 (s, 2H). ^13^C NMR (125 MHz, DMSO-d_6_) δ 163.6, 160.9, 159.9, 158.9, 139.8, 136.8, 135.6, 134.1, 133.9, 132.7, 131.7, 130.9, 129.6, 128.8, 128.4, 127.6, 127.4, 127.0, 126.7, 126.4, 126.4, 125.6, 123.8, 120.1, 119.3, 116.7, 108.4, and 34.0. IR (ATR, cm^−1^): (3300–2200, wide NH amide signal), 3268 (NH), 3171, 3050, 2888, 1669 (C=O), 1569, 1415, and 797. EI MS (70eV): *m*/*z* (%): 520 (M^+^, 7), 328 (100), 296 (20), 192 (12), 151 (17), and 148 (41). HRMS (ESI-QTOF) M + H calc. for C_30_H_21_ClN_4_OS: 521.1197, found: 521.1200.

##### 6-Methoxy-3-(((4-((4-(naphthalen-2-yl)pyrimidin-2-yl)amino)phenyl)thio) methyl)quinolin-2(1*H*)-one (**30c**)

Yellowish Solid (93%) M.p. 533–536 K. Rf Hex:AcOEt (4:6): 0.13. ^1^H NMR (400 MHz, DMSO-d_6_) δ 11.79 (s, 1H), 9.84 (s, 1H), 8.75 (s, 1H), 8.59 (d, *J* = 5.2 Hz, 1H), 8.26 (dd, *J* = 8.5, 1.8 Hz, 1H), 8.09–8.01 (m, 2H), 7.99–7.96 (m, 1H), 7.85 (d, *J* = 8.5 Hz, 2H), 7.64 (s, 1H), 7.62–7.57 (m, 2H), 7.56 (d, *J* = 5.2 Hz, 1H), 7.37 (d, *J* = 8.5 Hz, 2H), 7.23 (d, *J* = 8.8 Hz, 1H), 7.10–7.06 (m, 2H), 4.01 (s, 2H), and 3.68 (s, 3H). ^13^C NMR (100 MHz, DMSO-d_6_) δ 163.5, 160.8, 160.0, 159.0, 154.1, 139.7, 136.5, 134.1, 134.0, 132.7, 132.6, 131.3, 129.9, 128.9, 128.5, 127.6, 127.5, 127.0, 126.8, 126.7, 123.9, 119.6, 119.4, 119.0, 116.1, 108.9, 108.4, 55.3, and 33.9. IR (ATR, cm^−1^): (3400–2400, wide NH amide signal), 3270 (NH), 3172, 3002, 2832, 2362, 1665 (C=O), 1569, 1414, and 800. EI MS (70eV): *m*/*z* (%): 516 (M^+^, <0.3), 328 (100), 296 (22), and 148 (86). HRMS (ESI-QTOF) M + H calc. for C_31_H_24_N_4_O_2_S: 517.1693, found: 517.1698.

##### 3-(((4-((4-(Benzo[d][1,3]dioxol-5-yl)pyrimidin-2-yl)amino)phenyl)thio)methyl)quinolin-2(1*H*)-one (**31a**)

White Solid (74%) M.p. 567–570 K. Rf DCM:MeOH (9:1): 0.65. ^1^H NMR (400 MHz, DMSO-d_6_) δ 11.88 (s, 1H), 9.69 (s, 1H), 8.47 (d, *J* = 5.2 Hz, 1H), 7.78–7.74 (m, 3H), 7.70 (ps, 1H), 7.65 (s, 1H), 7.52 (d, *J* = 8.0 Hz, 1H), 7.43 (pt, *J* = 7.7 Hz, 1H), 7.37–7.31 (m, 3H), 7.28 (d, *J* = 8.4 Hz, 1H), 7.10 (pt, *J* = 7.7 Hz, 1H), 7.06 (d, *J* = 8.1 Hz, 1H), 6.12 (s, 2H), and 3.99 (s, 2H). ^13^C NMR (100 MHz, DMSO-d_6_) δ 163.0, 161.2, 159.8, 158.7, 149.7, 148.0, 139.7, 138.1, 136.8, 131.5, 130.7, 129.8, 129.5, 127.5, 126.5, 121.8, 121.7, 119.2, 119.0, 114.8, 108.5, 107.6, 106.7, 101.7, and 33.9. IR (ATR, cm^−1^): (3400–2400, wide NH amide signal), 3267 (NH), 3174, 3003, 2900, 2362, 1665 (C=O), 1571, 1413, and 799. EI MS (70eV): *m*/*z* (%): 480 (M^+^, 13), 322 (100), 158 (62), and 130 (35). HRMS (ESI-QTOF) M + H calc. for C_27_H_20_N_4_O_3_S: 481.1329, found: 481.1331.

##### 3-(((4-((4-(Benzo[d][1,3]dioxol-5-yl)pyrimidin-2-yl)amino)phenyl)thio)methyl)-6-chloroquinolin-2(1*H*)-one (**31b**)

White Solid (91%) M.p. 526–529 K. Rf Hex:AcOEt (4:6): 0.31. ^1^H NMR (400 MHz, DMSO-d_6_) δ 12.01 (s, 1H), 9.70 (s, 1H), 8.47 (d, *J* = 5.2 Hz, 1H), 7.77 (d, *J* = 8.5 Hz, 2H), 7.74 (dd, *J* = 8.1, 1.8 Hz, 1H), 7.69 (d, *J* = 1.8 Hz, 1H), 7.64 (d, *J* = 2.4 Hz, 1H), 7.62 (s, 1H), 7.46 (dd, *J* = 8.8, 2.4 Hz, 1H), 7.35–7.30 (m, 3H), 7.28 (d, *J* = 8.8 Hz, 1H), 7.05 (d, *J* = 8.1 Hz, 1H), 6.12 (s, 2H), and 3.97 (s, 2H). ^13^C NMR (100 MHz, DMSO-d_6_) δ 163.0, 161.0, 159.8, 158.7, 149.7, 148.0, 139.9, 136.8, 135.6, 131.7, 130.9, 130.7, 129.7, 126.4, 126.3, 125.7, 121.7, 120.1, 119.3, 116.7, 108.5, 107.6, 106.7, 101.7, and 34.0. IR (ATR, cm^−1^): (3400–2400, wide NH amide signal), 3266 (NH), 3178, 3001, 2828, 2362, 1668 (C=O), 1572, 1414, and 794. EI MS (70eV): *m*/*z* (%): 514 (M^+^, 1), 322 (100), 193 (25), and 145 (45). HRMS (ESI-QTOF) M + H calc. for C_27_H_19_ClN_4_O_3_S: 515.0939, found: 515.0935.

##### 3-(((4-((4-(Benzo[d][1,3]dioxol-5-yl)pyrimidin-2-yl)amino)phenyl)thio)methyl)-6-methoxyquinolin-2(1*H*)-one (**31c**)

Pale yellow Solid (94%) M.p. 515–518 K. Rf Hex:AcOEt (4:6): 0.14. ^1^H NMR (400 MHz, DMSO-d_6_) δ 11.78 (s, 1H), 9.69 (s, 1H), 8.47 (d, *J* = 5.2 Hz, 1H), 7.77 (d, *J* = 8.5 Hz, 2H), 7.74 (dd, *J* = 8.1, 1.8 Hz, 1H), 7.69 (d, *J* = 1.8 Hz, 1H), 7.62 (s, 1H), 7.34–7.32 (m, 3H), 7.22 (d, *J* = 8.8 Hz, 1H), 7.11–7.02 (m, 3H), 6.12 (s, 2H), 3.99 (s, 2H), and 3.70 (s, 3H). ^13^C NMR (100 MHz, DMSO-d_6_) δ 163.0, 160.8, 159.8, 158.7, 154.1, 149.7, 148.0, 139.7, 136.5, 132.6, 131.3, 131.3, 130.7, 129.9, 126.7, 121.7, 119.6, 119.3, 119.3, 119.0, 116.1, 108.9, 108.5, 107.6, 106.7, 101.7, 55.3, and 33.9. IR (ATR, cm^−1^): (3300–2400, wide NH amide signal), 3258 (NH), 3169, 2998, 2362, 1670 (C=O), 1572, 1420, and 796. EI MS (70eV): *m*/*z* (%): 510 (M^+^, 5), 322 (100), 188 (49), 145 (46), and 117 (22). HRMS (ESI-QTOF) M + H calc. for C_28_H_22_N_4_O_4_S: 511.1435, found: 511.1438.

##### 3-(((3-((4-(Naphthalen-2-yl)pyrimidin-2-yl)amino)phenyl)thio)methyl)quinolin-2(1*H*)-one (**35a**)

White Solid (80%) M.p. 534–537 K. Rf Hex:AcOEt (1:1): 0.21. ^1^H NMR (400 MHz, DMSO-d_6_) δ 11.39 (s, 1H), 9.27 (s, 1H), 8.71 (d, *J* = 1.8 Hz, 1H), 8.55 (d, *J* = 5.2 Hz, 1H), 8.24 (dd, *J* = 8.5, 1.8 Hz, 1H), 8.10 (t, *J* = 2.0 Hz, 1H), 8.06–8.01 (m, 2H), 7.97–7.94 (m, 1H), 7.79 (s, 1H), 7.66 (ddd, *J* = 8.2, 2.0, 1.0 Hz, 1H), 7.61–7.54 (m, 2H), 7.51–7.46 (m, 2H), 7.41 (ddd, *J* = 8.2, 7.2, 1.4 Hz, 1H), 7.32 (pd, *J* = 8.2 Hz, 1H), 7.27 (pt, *J* = 8.0 Hz, 1H), 7.09 (ddd, *J* = 8.2, 7.2, 1.4 Hz, 1H), 7.04 (ddd, *J* = 7.8, 2.0, 1.0 Hz, 1H), and 4.16 (s, 2H). ^13^C NMR (100 MHz, DMSO-d_6_) δ 163.3, 160.5, 159.6, 158.0, 140.6, 137.7, 136.3, 135.8, 133.7, 133.6, 132.3, 129.0, 128.2, 128.1, 127.7, 126.9, 126.7, 126.6, 126.4, 125.9, 123.3, 121.9, 121.0, 119.5, 118.5, 116.8, 114.3, 107.9, and 32.2. IR (ATR, cm^−1^): (3400–2400, wide NH amide signal), 3057, 2947, 2886, 1638 (C=O), 1584, 1214, and 747. EI MS (70eV): *m*/*z* (%): 486 (M+, 87), 453 (60), 328 (43), 158 (100), and 130 (43). HRMS (ESI-QTOF) M + H calc. for C_30_H_22_N_4_OS: 487.1587 found: 487.1585.

##### 6-Chloro-3-(((3-((4-(naphthalen-2-yl)pyrimidin-2-yl)amino)phenyl)thio)me-thyl)quinolin-2(1*H*)-one (**35b**)

White Solid (55%) M.p. 544–547 K. Rf DCM:MeOH (9:1): 0.22. ^1^H NMR (500 MHz, DMSO-d_6_) δ 12.03 (s, 1H), 9.81 (s, 1H), 8.76 (s, 1H), 8.57 (d, *J* = 5.2 Hz, 1H), 8.27 (dd, *J* = 8.5, 1.8 Hz, 1H), 8.10 (ps, 1H), 8.07–8.04 (m, 2H), 8.01–7.95 (m, 1H), 7.81 (s, 1H), 7.65–7.57 (m, 5H), 7.45 (dd, *J* = 8.8, 2.4 Hz, 1H), 7.30–7.25 (m, 2H), 6.99 (d, *J* = 7.8 Hz, 1H), and 4.11 (s, 2H). ^13^C NMR (125 MHz, DMSO-d_6_) δ 163.5, 161.0, 159.9, 158.9, 141.2, 136.8, 136.0, 135.9, 134.1, 133.9, 132.7, 130.6, 129.7, 129.2, 128.9, 128.4, 127.6, 127.5, 127.1, 126.7, 126.4, 125.7, 123.9, 121.7, 120.1, 118.9, 116.8, 116.7, 108.5, and 32.3. IR (ATR, cm^−1^): (3300–2400, wide NH amide signal), 3090, 2888, 1637 (C=O), 1584, 1290, and 772. EI MS (70eV): *m*/*z* (%): 520 (M+, 94), 487 (54), 328 (100), and 192 (56). HRMS (ESI-QTOF) M + H calc. for C_30_H_21_ClN_4_OS: 521.1197 found: 521.1195.

##### 6-Methoxy-3-(((3-((4-(naphthalen-2-yl)pyrimidin-2-yl)amino)phenyl)thio) methyl)quinolin-2(1*H*)-one (**35c**)

White Solid (87%) M.p. 534–537 K. Rf Hex:AcOEt (1:1): 0.07. ^1^H NMR (500 MHz, DMSO-d_6_) δ 11.82 (s, 1H), 9.92 (s, 1H), 8.77 (s, 1H), 8.58 (d, *J* = 5.2 Hz, 1H), 8.26 (dd, *J* = 8.5, 1.8 Hz, 1H), 8.10 (ps, 1H), 8.07–8.03 (m, 2H), 7.97 (dd, *J* = 7.8, 1.8 Hz, 1H), 7.82 (s, 1H), 7.64–7.54 (m, 4H), 7.29 (pt, *J* = 8.0 Hz, 1H), 7.22 (d, *J* = 8.8 Hz, 1H), 7.09–7.04 (m, 2H), 7.00 (pd, *J* = 7.8 Hz, 1H), 4.13 (s, 2H), and 3.70 (s, 3H). ^13^C NMR (120 MHz, DMSO-d_6_) δ 163.9, 160.8, 159.5, 158.3, 154.2, 140.9, 136.7, 136.4, 134.2, 133.8, 132.7, 132.7, 129.5, 129.2, 128.9, 128.5, 127.7, 127.6, 127.3, 126.8, 123.9, 121.5, 119.6, 119.2, 118.7, 116.8, 116.2, 108.9, 108.4, 55.4, and 32.1. IR (ATR, cm^−1^): (3300–2200, wide NH amide signal), 3165, 3056, 1623 (C=O), 1584, 1231, and 769. EI MS (70eV): *m*/*z* (%): 516 (M+, 39), 483 (25), 188 (100), and 160 (18). HRMS (ESI-QTOF) M + H calc. for C_31_H_24_N_4_O_2_S: 517.1693 found: 517.1691.

### 3.3. Enzymatic Assay

*h*LDHA activity was determined throughout a fluorometric method with pyruvate as substrate and NADH co-factor, as previously reported [66,68,69] and modified as described here: in each well, the final volume was set to 200 µL, and the final concentrations were 100 mM potassium phosphate buffer, 0.041 units/mL *h*LDHA (95%, specific activity >300 units/mg and concentration of 0.5 mg/mL, Abcam, Cambridge, United Kingdom), 151 µM β-NADH, 1 mM pyruvate (saturated conditions), and DMSO solutions (2%, *v*/*v*) of pure compounds at concentrations in the range of 0.048–100 µM. The reaction was initiated by the addition of pyruvate, and the NADH concentration decrease was measured for 10 min in a TECAN Infinite 200 Pro M Plex fluorescence plate reader at 28 °C, with excitation at 340 nm and emission at 460 nm. The percentage of activity for each measure was calculated by comparison between the maximum slope of each compound concentration and the maximum slope when no inhibitor (DMSO only) was in the well (100% enzymatic activity). The compound 3-[[3[(cyclopropylamino)sulfonyl]-7-(2,4-dimethoxy-5-pyrimidinyl)-4-quinolinyl]amino]-5-(3,5-difluorophenoxy)benzoic acid (GSK 2837808 A, Tocris, Minneapolis, MN, USA) was used as a positive control [52].

The measurements were obtained thrice, and data were expressed as the mean ± SD of *n* = 3 replicates for IC_50_ values. As-obtained data were later plotted in GraphPad Prism version 5.00 for Windows (GraphPad Software, La Jolla, CA, USA). Nonlinear regression analysis was chosen for dose response curve, representing the logarithm of inhibitor concentration vs. normalized enzymatic activity in order to calculate IC_50_ values. For individual dose–response inhibition curve of hybrids having IC_50_ < 100 μM, see Appendix A.

### 3.4. Molecular Modeling

The molecular modeling and Docking analysis were performed using the MOE 2020.09 suit from Chemical Computing Group’s Molecular Operating Environment, and the minimization of the energy of molecules and complexes were performed under molecular mechanics using the Amber14:EHT force field.

The complex of the *h*LDHA protein with the inhibitor **W31**, with PDB code **4R68**, was downloaded from the Protein Data Bank (PDB) and prepared as follows: all the chains but one were deleted using the sequence editor (SEQ), the hydrogens were added to structure with the “Protonate 3D” tool and checked for the right charge in any heteroatom, and finally, the complex system was minimized using the force field Amber14. The energy minimization mode used is named “General”, in which force field minimization is performed with emphasis on tether layers. No restraints are applied. Constraints selected were to maintain rigid water molecules. The gradient was of 0.1 RMS, meaning that the energy minimization was finished when the root means square gradient fell below the specified value (0.1).

The input database of screened molecules were prepared from builder editor and imported in the corresponding database file (*.mdb), which was used as the input file in the docking process. To prepare the database input file, we followed a similar preparation process that included a first wash (set of cleaning rules to ensure that each structure is in a suitable form for subsequent modelling steps, such as conformational enumeration and protein-ligand docking), checking for the right partial charges, and finally, minimizing the energy of the molecules using the force field Amber14.

Three pharmacophoric models were created from the Pharmacophore Query Editor tool: (i) **W31** site, (ii) NADH site, and (iii) extended site w31-NADH site. Three features were defined so as to interact with the main amino acid residues: Asn^137^, Arg^168^, His^192^, and Asp^194^. All three features were defined with a radius of 1.2 Å, and none of them was classified as essential nor ignored. When stablishing the search criteria, the partial match was clicked on and defined as at least 1 interaction with one of those features.

The docking screening was carried out with the following settings: Receptor: MOE (the previously prepared complex), receptor atoms; Site: Ligand atoms: Wall constraint: on; Pharmacophore: on; Ligand: MDB file (the input *.mdb database); Placement: Pharmacophore; Number of returned poses (poses returned by each ligand’s placement): 3000; Placement score: London dG; Placement poses: 100; Refinement method: rigid receptor; Refinement score: GBI/WSA dG; Refinement poses (number of poses retained to be written in the output file): 10.

Once the docking was complete, the best pose score for each ligand determined by a further minimization process (in the output file) was required using molecular mechanics and the specified forcefield. The best pose was determined by the following criteria: (i) RMSD [64] < 1.8 Å, (ii) affinity (S) [65] values < −9 kcal/mol, and (iii) energy values involved in the interactions with the main amino acid residues [65], selecting those interacting with Arg^168^ firstly and afterwards those with the higher number of interactions. In the case that they all interacted with the same amino acids, the ones with the highest energy values involved in the interactions with those amino acid residues were chosen.

## 4. Conclusions

After having synthesized and evaluated a first set of pyrimidine-quinolone hybrids, due to the different reasons explained, we designed, synthesized, and evaluated novel *h*LDHA inhibitors 1,2-linked (**24**–**27**(**a**–**c**)), 1,3-linked (**35**(**a**–**c**)), and 1,4-linked (**28**–**31**(**a**–**c**)) pyrimidine-quinolone hybrids. Molecular modelling (docking) predicted that hybrids 1,2-linked were the most interesting ones to inhibit the *h*LDHA enzyme and that the 1,4-linked ones were inactive. Additionally, those hybrids having the naphthalene-2-yl moiety as the hydrophobic structure were predicted to be the most interesting ones.

Enzymatic assays confirmed the in silico predictions and a preliminary SAR was established, and 1,3-linked hybrids **33**–**36**(**a**–**c**) were included for the study.

Data from SAR analysis enabled us to explain the difference in the experimental IC_50_ values between the different U-shaped pyrimidine-quinolone hybrids and predicted those 1,3-linked hybrids to have an intermediate inhibitory activity between those 1,2- and 1,4-linked, with a bias towards the U-shaped ones. In this way, hybrids **35**(**a**–**c**) with the naphthalene-2-yl moiety were synthesized and evaluated, confirming the predictions from SAR analysis.

In summary, we have been able to design and synthesize a new family of *h*LDHA inhibitors with good IC_50_ values and designed a preliminary SAR, which encourages us to design a promising next generation in order to improve their inhibitory potency.

## Data Availability

Data is contained within the article and Appendix A.

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
