# Peer review of "Design and Synthesis of New Pyrimidine-Quinolone Hybrids as Novel hLDHA Inhibitors"

_pharmaceuticals, 2022, doi:10.3390/ph15070792_

Round 1

Reviewer 1 Report

The work is continuation of previous works of the authors. The methods are well established and the rationale for the research is good. Therefore - poor novelty, but this is typical case in continuation of the studies.

Author Response

The whole manuscript was revised again by all the authors and a native English speaker teacher (who is now acknowledged in the MS) to check for spellings, grammar errors or misunderstandings. And changes were done using the control task tool in MS word.

Reviewer 2 Report

The authors synthesized a lot of new pyrimidine-Quinoline compounds and tested the against hLDHA inhibitor's activity. In that, they identified 25a, 25b, and 35a active compounds. The manuscript was well written in a good way to understand the readers.

Minor:

1.     I suggest to the authors add figure on the first page, helping the readers to understand the background of this work.

2.     The figure 7 was missed in the manuscript, please include that.

Author Response

Answer to query 1: Figure 3 was rebuilt to show the background of this work showing the step-path for the synthesis of hybrids in the previous works as SPhK1 and P-gp inhibitors and in the present work, remarking the different synthetic approaches to reach the final hybrid compounds; We do not know if this was exactly the referee's suggestion. To put this at the beginning on intro would change the whole structure of the introduction part and need major changes.

Answer to query 2: if the docx file for the submission is downloaded the referee can see figure 7 in it. Maybe he was using the pdf, that sometimes take out some figures, in its conversion from docx file

Reviewer 3 Report

This paper aims to discuss the design and synthesis of new pyrimidine-quinolone hybrids as novel hLDHA inhibitors.

The manuscript is written comprehensively enough to be understandable.

The paper stated the purpose, discussion and global implication are clearly stated and consistent with the rest of the manuscript; Authors provided enough information in their discussion by using a good number of important articles talked about the subject. They also explained their attempts to get some target molecules identifying why they were unsuccessful to get them.

The authors clearly identified some examples of pyrimidine and quinolones derivatives designed and synthesized previously and they clarified their work purpose.

The authors addressed their hypothesis and opinion in a reproducible way and proved their results through all the required experiments and analysis, they used enough number of analyses to prove their results and their SAR study. The results were presented in a clear way which facilitate in reaching a conclusion.

I recommend to talk more about quinolone derivatives and pyrimidine derivatives separately elucidating the positive effect of hybridisation of both molecules in term of biological activity in the introduction supported by good number of references from the literature, I recommend   these papers :

Structure activity relationships and the binding mode of quinolinone-pyrimidine hybrids as reversal agents of multidrug resistance mediated by P-gp

Scientific Reports volume 11, Article number: 16856 (2021

Quinoline-Based Hybrid Compounds with Antimalarial Activity

Molecules. 2017 Dec; 22(12): 2268. doi: 10.3390/molecules22122268

Preliminary Studies of Antimicrobial Activity of New Synthesized Hybrids of 2-Thiohydantoin and 2-Quinolone Derivatives Activated with Blue Light

Molecules 2022, 27, 1069. https://doi.org/10.3390/molecules27031069

No plagiarism has been detected.

Figure 7 is missing

385: Scheme 3 & 4: The temperature is missing in the 1st reaction

References: some references are missing doi: 5, 9, 26 and 63

Author Response

The whole manuscript was revised again by all the authors and a native English speaker teacher (who is now acknowledged in the MS) to check for spellings, grammar errors or misunderstandings. And changes were done using the control task tool in MS word.

Answers to query 1 relative to references in introduction:

  1. Structure activity relationships and the binding mode of quinolinone-pyrimidine hybrids as reversal agents of multidrug resistance mediated by P-gp. Scientific Reports volume 11, 16856 (2021). Answer: this reference was already included previously in reference 59, and also in figure 3; reference was also fixed including now the volume.
  2. Quinoline-Based Hybrid Compounds with Antimalarial Activity, Molecules. 2017 Dec; 22(12): 2268.  doi: 10.3390/molecules22122268, Answer: this was included as new reference [26] line 55 as example for antimalarial.
  3. 3. Preliminary Studies of Antimicrobial Activity of New Synthesized Hybrids of 2-Thiohydantoin and 2-Quinolone Derivatives Activated with Blue Light, Molecules 2022, 27, 1069. https://doi.org/10.3390/molecules27031069, Answer: the hybridization with different scaffold was indicated, and each scaffold was named separately in lines 57-59, including the hydantoin -quinoline hybrids related to this reference suggested by the referee
  • Query 2: Figure 7 is missing; Answer: Figure 7 appears in the docx file that I just downloaded but not in the corresponding pdf. That maybe came from the conversion from doc files that sometimes take out some figures. If Referee is using the pdf that is the problem.
  • Query 3,  385: Scheme 3 & 4: The temperature is missing in the 1st reaction; Answer: the temperature for the first step in both schemes was room temperature, which (r.t.)  was included in both as suggested,
  • Query 4: References: some references are missing doi: 5, 9, 26 and 63; Answer: It seems to be a problem with the Mendeley update of the doi information; all of them have now their corresponding doi. As the refresh tool or update of references in word is not shown in the control task, I have underlined in the doc version with control task to get it clear.